# Efflux pump-mediated resistance to antifungal compounds can be prevented by conjugation with triphenylphosphonium cation

Wenqiang Chang[1], Jun Liu[1], Ming Zhang[1], Hongzhuo Shi[1], Sha Zheng[1], Xueyang Jin[1], Yanhui Gao[1], Shuqi Wang[1], Aiguo Ji[1] & Hongxiang Lou[1]

Antifungal resistance due to upregulation of efflux pumps is prevalent in clinical *Candida* isolates. Potential efflux pump substrates (PEPSs), which are active against strains deficient in efflux pumps but inactive against wild-type strains, are usually missed in routine antifungal screening. Here we present a method for identification of PEPSs, and show that conjugation with mitochondria-targeting triphenylphosphonium cation (TPP$^+$) can enhance or restore the compounds' antifungal activity. The screening method involves co-culturing a wild-type *C. albicans* strain and a Cdr efflux pump-deficient strain, labelled with different fluorescent proteins. We identify several PEPSs from a library of natural terpenes, and restore their antifungal activity against wild-type and azole-resistant *C. albicans* by conjugation with TPP$^+$. The most active conjugate (IS-2-Pi-TPP) kills *C. albicans* cells, prevents biofilm formation and eliminates preformed biofilms, without inducing significant resistance. The antifungal activity is accompanied by mitochondrial dysfunction and increased levels of intracellular reactive oxygen species. In addition, IS-2-Pi-TPP is effective against *C. albicans* in a mouse model of skin infection.

[1] Key Lab of Chemical Biology of Ministry of Education, School of Pharmaceutical Sciences, Shandong University, Jinan 250012, China. These authors contributed equally: Wenqiang Chang, Jun Liu.  Correspondence and requests for materials should be addressed to H.L. (email: louhongxiang@sdu.edu.cn)

Candida species are among the most common clinical etiologies of fungal infections and cause high morbidity and mortality rates[1–3]. The prevalence of antifungal resistance due to the prolonged use of fungistatic drugs has become a challenge in the treatment of fungal infections, highlighting the urgent need for developments in the antifungal pipeline[4–7]. Among the variety of molecular mechanisms of fungal resistance that *Candida* species have evolved[8,9], the upregulation of the efflux pumps *MDR1* and/or *CDR1/CDR2* contributes to azole resistance in most clinical *Candida albicans* isolates[10,11]. *MDR1* and *CDR1/CDR2* expression are regulated by the transcription factors Mrr1 and Tac1, respectively. Gain-of-function mutations in Mrr1 or Tac1 result in constitutive over-expression of *MDR1* or *CDR1/CDR2*, respectively, and increase azole resistance[12,13].

Natural products have long been regarded as potential sources for the discovery of novel drug leads[14,15]. In our continuing research on varied natural products with antifungal activities from bryophytes and endolichenic fungi, we found that several sesquiterpenes isolated from Chinese liverworts were active against efflux pump-deficient *C. albicans* strains but inert towards wild-type strains[16,17]. These terpenes are therefore potential efflux pump substrates (PEPSs), which cannot be intracellularly retained in the wild-type strain due to efflux by efflux pumps.

Co-culture assays in which an antibiotic-sensitive strain is mixed with an antibiotic-resistant one have been used to elucidate mechanisms of resistance or discover antibacterial agents that can select against resistant strains due to "collateral sensitivity" or inversion of resistance[18,19]. Co-culture assays in which a wild-type strain and efflux pump-deficient strain are mixed can be used for high-throughput screening to simultaneously identify active agents against the wild-type strain and PEPSs.

Strategies for restoring the activity of PEPSs against wild-type strains include adding efflux pump inhibitors or modifying the chemical structures of PEPSs to prevent their efflux by Cdrs[16,20]. However, these strategies are associated with some disadvantages, such as lower selectivity and reduced antifungal activity. Developing alternative strategies for avoiding efflux by pumps will expand opportunities to discover antifungal agents from natural products. Bacterial cells can synergistically decrease the permeability of cytoplasmic membrane barriers and increase active efflux to prevent intracellular access to toxic agents and acquire resistance[21]. Cell-penetrating peptides or lipophilic cations can facilitate the delivery of membrane-impermeable molecules into living cells[22,23]. Triphenylphosphonium cation (TPP$^+$), a lipophilic cation, can easily pass through lipid bilayers due to its high lipophilicity and stable cationic charge, and the potential gradient drives the accumulation of TPP$^+$-conjugates in the mitochondrial matrix. Conjugation of TPP$^+$ to anticancer drugs has been used to improve their anticancer efficacy by increasing the drug distribution in the mitochondria, including an attempt in our previous study[24–28].

In this study, we establish an assay in which a wild-type *C. albicans* strain and a Cdr-deficient strain are co-cultured to screen a natural terpene library and identify several PEPSs. We restore the antifungal activity of these PEPSs against wild-type *Candida* strains by chemical conjugation with TPP$^+$. These TPP$^+$-conjugates can pass through the cytoplasmic membrane, bypass active export by efflux transporters and accumulate within the mitochondria to kill *Candida* cells. The screening assay established in this study can be used to mine masked potentially active agents from natural products, and TPP$^+$ conjugation provides an alternative approach for converting PEPSs into active antifungal agents. The strategy used in this study can help increase the development of antifungal drugs from natural products that are usually ignored in traditional bioassays and address the drug-resistance problems arising from efflux pumps.

## Results

**Hits against an efflux pump-deficient strain**. Traditional antifungal screening using wild-type strains often fails to discover PEPSs. In this study, we established a screening assay in which equal quantities of two *C. albicans* strains labeled with different fluorescent proteins (FPs) were co-cultured. One strain was the wild-type isolate *TDH3-RFP*-CAI4, which was labeled with red fluorescent protein (RFP) as a fusion with the Tdh3 protein (Tdh3p) in strain CAI4 and emitted strong red fluorescence due to the high expression of Tdh3p in *C. albicans*. The other was the efflux pump-deficient strain *TDH3-GFP*-DSY654, which was labeled with green fluorescent protein (GFP) as a fusion with Tdh3p in strain DSY654 (*cdr1* and *cdr2* double mutant) (Fig. 1a). The assay was performed in RPMI1640 liquid medium using equal mixtures of the two strains cultured for 48 h with a linear dilution series of each tested compound. The selective inhibitory effects of the compounds on the wild-type strain and efflux pump-deficient strain were evaluated using a multifunctional plate reader to measure the fluorescence intensities of GFP and RFP. We utilized this assay to screen a small terpene library containing 170 bryophyte-derived compounds and two semi-synthesized *ent*-beyerene diterpenoids (Supplementary Figure 1) as a representative collection of the structural and biological diversity of terpenes[29]. Five hits that were active against both the wild-type strain and efflux pump-deficient strain were obtained, and 18 PEPSs were detected (Supplementary Figure 2, Fig. 1b and c). These results were further confirmed by testing the minimum inhibitory concentrations (MICs) of the compounds against the individual strains (Fig. 1c). Thus, this co-culture assay could be used for high-throughput screening for hits in one step to increase the discovery of antifungal agents from natural products.

**TPP$^+$ conjugation restores the antifungal activity of PEPSs**. Inspired by previous results, we theorized that an approach that allowed the PEPSs to be carried into cells and avoid efflux by pumps would convert the PEPSs into antibiotics with activity against wild-type strains. TPP$^+$ can easily pass through lipid bilayers and has been conjugated with anticancer drugs to improve anticancer efficacy by increasing drug distribution into the mitochondria[24–28]. We therefore assessed whether TPP$^+$ could carry PEPSs into *C. albicans* cells and avoid efflux by pumps. One of the PEPSs, the monoterpenoid 2α,5β-dihydroxybornane-2-cinnamate (DBC), has a chemically modifiable OH group. When DBC was conjugated with TPP$^+$, the resultant product (DBC-TPP) (Supplementary Figure 3) exhibited potent activity against wild-type *C. albicans* strain SC5314, which features a low level of expression of efflux pump-encoding genes under non-inducing conditions. DBC-TPP was similarly effective against two other strains, YEM13 and YEM15, with upregulated expression of the efflux pump-encoding genes *MDR1* and *CDR1/CDR2*. By contrast, the parent compounds DBC and methyltriphenylphosphonium bromide were inert towards the tested *C. albicans* isolates (Table 1 and Supplementary Table 1). (+)-Borneol, which has a core structure similar to that of DBC, also harbored antifungal activity when conjugated with TPP$^+$ (Table 1, Supplementary Table 1 and Supplementary Figure 4). The PEPSs also included several *ent*-kaurane diterpenoids (Fig. 1c), but their limited quantities prevented their direct modification. Consequently, isosteviol (IS-1), which is easily accessed by the hydrolysis of stevioside, and its analog IS-2 were conjugated with TPP$^+$ (Fig. 2a and Supplementary Figures 5–7).

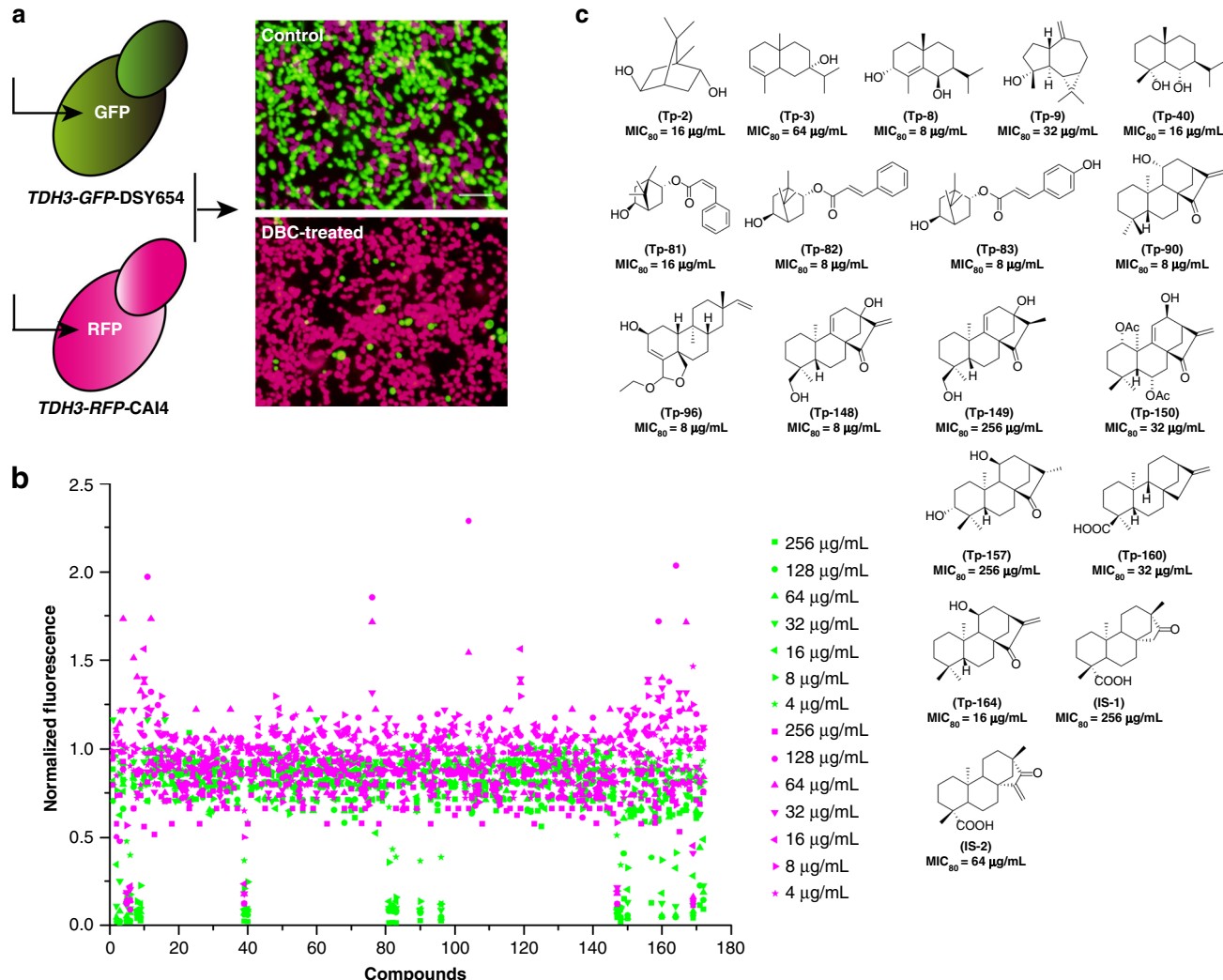

**Fig. 1** Identification of PEPSs by screening a terpene library. **a** The *C. albicans* wild-type strain *TDH3-RFP*-CAI4 (shown in magenta) and Cdr-deficient strain *TDH3-GFP*-DSY654 (shown in green) were constructed for the co-culture assay. DBC, a representative PEPS, was used to demonstrate the validity of this assay. **b** Two fluorescently labeled strains were equally mixed and exposed to two-fold serial dilutions of 172 terpenes ranging from 256 to 2 μg/mL for 48 h at 30 °C. The GFP and RFP fluorescence intensities were separately detected using a microplate reader. Agents that reduced either GFP or RFP fluorescence intensity by more than 50% compared with the control group were characterized as hits. **c** The chemical structures of the PEPSs and their MIC values against the Cdr-deficient strain

The resultant TPP$^+$-conjugates, IS-1-TPP and IS-2-TPP, were also active against the wild-type *C. albicans* strains (Table 1 and Supplementary Table 1). Our previous study revealed that sola-sodine, a steroidal alkaloid, is another type of PEPS[18]. The TPP$^+$ conjugation strategy was also applied to convert solasodine into an agent (Sola-TPP) that was active against wild-type or efflux pump-activated *C. albicans* strains (Table 1, Supplementary Table 1 and Supplementary Figure 8).

In addition, we observed that these five TPP$^+$-conjugates were also active against other types of azole-resistant *C. albicans* strains. Gain-of-function mutations in Tac1, Mrr1, and Upc2 result in constitutive overexpression of *CDR1/CDR2*, *MDR1*, and *ERG11* and increase azole resistance in *C. albicans*[30–33]. Several constructed or clinical strains with gain-of-function mutations in Tac1, Mrr1, and Upc2 exhibited similar susceptibility to the tested TPP$^+$-conjugates (Supplementary Tables 2 and 3), suggesting that TPP$^+$-conjugates could address drug-resistance problems resulting from efflux pumps and other azole-resistance mechanisms.

**TPP$^+$ conjugation facilitates mitochondrial delivery of PEPSs.** To test whether the great discrepancy in bioactivity between the TPP$^+$-conjugates and their parent compounds was attributable to differences in intracellular accumulation, we measured the total intracellular contents of the tested drugs distributed in the mitochondria and other cellular compartments by high-performance liquid chromatography (HPLC). The average intracellular contents of TPP$^+$-conjugates in wild-type *C. albicans* strain SC5314 ranged from 5.8 to 32.32 μg per $10^8$ cells, whereas the amounts of the parent compounds were below the threshold of detection (Fig. 2b and Supplementary Table 4). Moreover, the TPP$^+$-conjugates also accumulated in the efflux pump-activated strains YEM13 and YEM15. The intracellular contents of the TPP$^+$-conjugates were slightly lower in YEM13 and YEM15 cells than in wild-type cells (Supplementary Table 4) but were suffi-cient to guarantee activity. These observations indicate that the parent compounds were expelled by efflux pumps; by contrast, TPP$^+$ facilitated their transport into cells and bypassing of active expulsion by efflux pumps. This conclusion was further

confirmed by an additional test that revealed accumulation of a (dansyl)-labeled TPP$^+$-conjugate (Ds-TPP) fluorescent probe in *C. albicans* cells but exclusion of dansyl chloride (Fig. 2c, Supplementary Figures 9 and 10). As a fluorescent probe, we also synthesized RhB-TPP (Supplementary Figure 11), a TPP$^+$ conjugate of rhodamine B (RhB), which is considered a fluorescent substrate of multixenobiotic transporters[34]. The intracellular contents of RhB-TPP in wild-type, efflux pump-hyperactivated and efflux pump-deficient strains were measured by detecting fluorescence intensity by flow cytometry. The intracellular contents of RhB were higher in the efflux pump mutant strain DSY654 than in the other strains, including wild-type or efflux pump-activated strains. Compared with RhB, the intracellular contents of RhB-TPP were notably higher and were maintained at similar levels in the different strains (Supplementary Figure 12), indicating minimal export of the TPP$^+$-conjugate by efflux pumps. Finally, we observed that the antifungal activity of the TPP$^+$-conjugates against efflux pump-deficient strains was similar to that against wild-type strains (Supplementary Table 5), further supporting the above conclusions.

TPP$^+$ is considered a mitochondria-targeting moiety capable of transporting active agents into the mitochondria[24–27]. To test whether the TPP$^+$-conjugates accumulated in fungal mitochondria, *C. albicans* cells were treated with DBC-TPP, borneol-TPP, IS-1-TPP, IS-2-TPP, and Sola-TPP. The mitochondria and other intracellular components were separately extracted, and the drug contents in these two fractions were detected using liquid chromatography–tandem mass spectrometry (LC–MS/MS). The results showed that the contents of the drugs in the mitochondria represented an average of 52–78% of the total cell contents (Fig. 2d), confirming the role of TPP$^+$ as a mitochondria-targeting moiety in fungal cells.

**The linker type affects the activity of TPP$^+$ derivatives**. Among the TPP$^+$-conjugates, IS-2-TPP displayed the most potent activity, probably due to the presence of an α,β-unsaturated ketone motif (Fig. 2a and Table 1). To obtain more active derivatives, several IS-2-TPP analogs with different linkers but the same α,β-unsaturated ketone core structure were synthesized. First, with the carboxylic acid functional group as the point of attachment, IS-2-TPP conjugates (**21–23, 28**) with hydrophobic polymethylene linkers were synthesized (Supplementary Figure 7). Next, linkers containing nitrogen atoms were designed. However, IS-2 tended to form dimers when aliphatic diamines were introduced into the linkers. Then, a piperazine group was introduced into the polymethylene linker, and conjugates (**33–38**) with different carbon chain lengths were synthesized (Supplementary Figure 7). Targeted amphiphilic TPP$^+$-conjugates (**44, 46, 49**) of IS-2 with PEG as a linker or a hydroxyl group on the linker were also synthesized (Supplementary Figure 13). The synthesized derivatives were characterized by nuclear magnetic resonance ($^1$H NMR, $^{13}$C NMR) and high-resolution mass spectrometry.

The antifungal activities of the resulting compounds against *C. albicans*, including the wild-type strain SC5314, the efflux pump-hyperactivated strains YEM13 and YEM15 and two clinical azole-resistant strains, were examined. The clinically used antifungal drugs fluconazole (FLC) and amphotericin B (AMB) were used as positive controls. The MIC assay revealed that the type and length of the linkers indeed affected activity; however, the variation in bioactivity was within 16-fold (Supplementary Table 6). The activity of the conjugates (**33–38**) with a piperazine group in the linker was higher than that of the conjugates with other types of linkers. The linker with a piperazine group and a 12-carbon chain length in compound IS-2-Pi-TPP (**35**) was optimal for antifungal

**Table 1 Antifungal activities of representative compounds**

| Compounds[a] | MIC[b,c] (µg/mL) | | | | |
|---|---|---|---|---|---|
| | SC5314 | YEM13 | YEM15 | 28A | 28I |
| DBC | >256 | >256 | >256 | >256 | >256 |
| DBC-TPP (**2**) | 4 | 4 | 4 | 4 | 4 |
| Borneol | >256 | >256 | >256 | >256 | >256 |
| Borneol-TPP (**3**) | 8 | 8 | 8 | 8 | 8 |
| IS-1 (**5**) | >256 | >256 | >256 | >256 | >256 |
| IS-1-TPP (**18**) | 8 | 8 | 8 | 8 | 8 |
| IS-2 (**11**) | >256 | >256 | >256 | >256 | >256 |
| IS-2-TPP (**22**) | 2 | 2 | 2 | 2 | 2 |
| IS-2-Pi-TPP (**35**) | 0.5 | 0.5 | 0.5 | 1 | 1 |
| Solasodine | >256 | >256 | >256 | >256 | >256 |
| Sola-TPP (**39**) | 4 | 4 | 4 | 4 | 4 |
| MTPPBr | >256 | >256 | >256 | >256 | >256 |
| Ds-TPP (**41**) | >256 | >256 | >256 | >256 | >256 |
| RhB-TPP (**42**) | >256 | >256 | >256 | >256 | >256 |
| FLC[b] | 2 | 64 | 64 | >128 | >128 |
| AMB | 1 | 0.5 | 2 | 2 | 2 |

[a]Sola-TPP is the TPP$^+$-conjugate of solasodine. MTPPBr denotes methyltriphenylphosphonium bromide. Ds-TPP is the (dansyl)-labeled TPP$^+$-conjugate. RhB-TPP is the TPP$^+$-conjugate of rhodamine B
[b]The MICs of FLC against these *C. albicans* isolates were reported in our previous publications
[c]SC5314 is the *C. albicans* wild-type strain; YEM13 is a constructed *C. albicans* strain with hyperexpression of *MDR1*; YEM15 is a constructed *C. albicans* strain with hyperexpression of both *CDR1* and *CDR2*; *C. albicans* 28A and 28I are clinical pan-azole-resistant strains

activity. Furthermore, the position of the piperazine group in the linker had no effect on activity.

Several active agents were chosen to test their activity against other *Candida* species, including *Candida krusei*, *Candida parapsilosis*, *Candida glabrata*, and *Candida tropicalis*. The agents also exhibited potent activity against these *Candida* species (Supplementary Tables 6 and 7).

**IS-2-Pi-TPP disrupts the fungal mitochondrial function**. IS-2-Pi-TPP, which features a piperazine-containing linker, displayed the highest fungicidal activity and was chosen for subsequent studies. The time-killing curves revealed that 4 µg/mL IS-2-Pi-TPP completely killed *C. albicans* cells within 4 h and that its activity was comparable to that of AMB (Supplementary Figure 14). Given the function of TPP$^+$ as a mitochondria-targeting moiety, we hypothesized that IS-2-Pi-TPP might disrupt the normal function of the mitochondria. By using the probe JC-1, we measured the mitochondrial membrane potential (MMP). Flow cytometry analysis revealed that IS-2-Pi-TPP remarkably increased the MMP of *C. albicans*, suggesting that the mitochondrial membrane was impaired (Fig. 3a). Tom70 is a major surface receptor for mitochondrial protein precursors in the outer mitochondrial membrane complex[35,36]. In this study, we used the previously constructed GFP-labeled *C. albicans* strain *TOM70-GFP-CAI4* to monitor the effect of IS-2-Pi-TPP on mitochondrial morphology. Confocal laser scanning microscopy (CLSM) observations based on Tom70-GFP imaging revealed that non-drug-treated cells had an extensive network of mitochondria displaying a tubular structure, whereas those in IS-2-Pi-TPP-treated cells presented an aggregated distribution and a disrupted tubular structure (Fig. 3b). When the normal function of the mitochondria is compromised, reactive oxygen species (ROS) generation tends to increase, which in turn damages the mitochondria and subsequently causes cell death[37,38]. Using MitoSOX Red, we probed mitochondrial ROS levels in the presence of IS-2-Pi-TPP. Flow cytometry analysis revealed that most of the IS-2-Pi-TPP-treated cells maintained high oxidative states and were

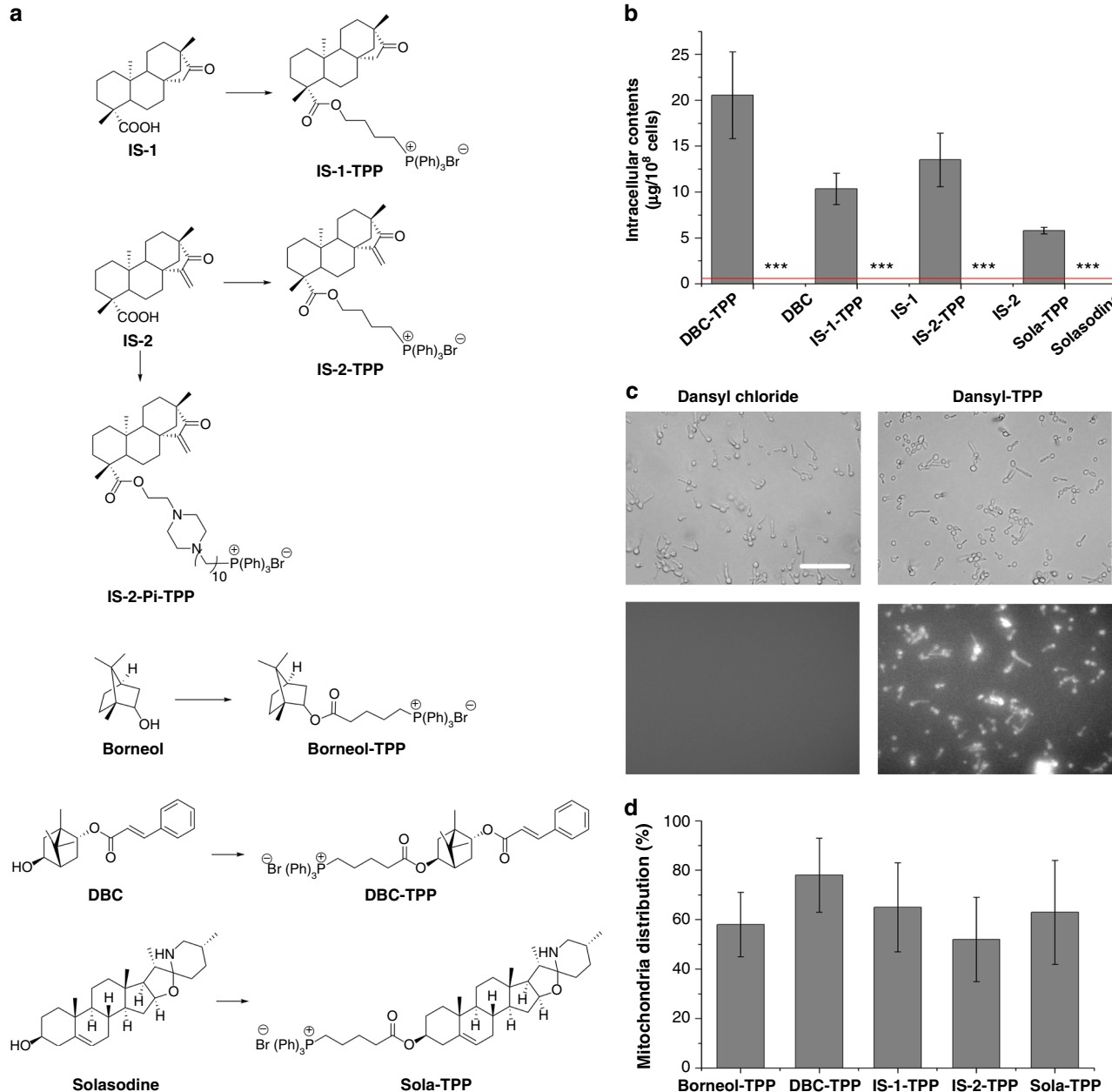

**Fig. 2** TPP+ targets PEPSs to the mitochondria of *C. albicans* cells. **a** The chemical structures of representative TPP+-conjugates and their parent compounds. **b** The intracellular contents of TPP+-conjugates and their parent compounds in the treated *C. albicans* cells were measured using HPLC. The detection threshold using HPLC was 0.1 μg for DBC and 0.4 μg for the other three parent compounds that was indicated by a red line. The HPLC analysis results demonstrated that these TPP+-conjugates accumulated in *C. albicans* cells, whereas their parent compounds were not transported into cells. Data are shown as means ± s.e.m. from three independent experiments. ***$P < 0.001$ for TPP+-conjugates versus their parent compounds by Student's *t*-test. **c** Fluorescence microscopy revealed that Ds-TPP accumulated in *C. albicans* cells cultured in RPMI1640 medium, whereas its parent compound dansyl chloride did not. Scale bar, 100 μm. **d** The accumulation of the representative tested compounds in *C. albicans* mitochondria. The contents of the tested compounds in the mitochondria and other intracellular components were determined by LC–MS/MS, and the ratio of the tested compounds in the mitochondria was calculated. Data are shown as means ± s.e.m. from three independent experiments

penetrated by the live cell-impermeable dye SYTOX (Fig. 3c), consistent with the CLSM observations (Fig. 3d). MMP alteration and increased ROS generation are considered characteristic features of cell apoptosis. We therefore detected cytochrome c (Cyt c) release from mitochondria. Compared with control cells, mitochondrial Cyt c levels decreased under IS-2-Pi-TPP treatment, whereas cytosolic levels increased significantly, indicating that IS-2-Pi-TPP induced the release of Cyt c from mitochondria

in *C. albicans* (Figs. 3e). 4′,6-Diamidino-2-phenylindole (DAPI) staining demonstrated that the chromatin appeared as single round spots in non-treated cells, whereas DAPI was dispersed in the whole cell in IS-2-Pi-TPP-treated cells (Fig. 3f). Annexin V-FITC and propidium iodide (PI) staining results revealed that most cells were stained with both Annexin V-FITC and PI under treatment with 2 μg/mL IS-2-Pi-TPP. Upon treatment with 4 μg/mL IS-2-Pi-TPP, the majority of cells were stained with PI

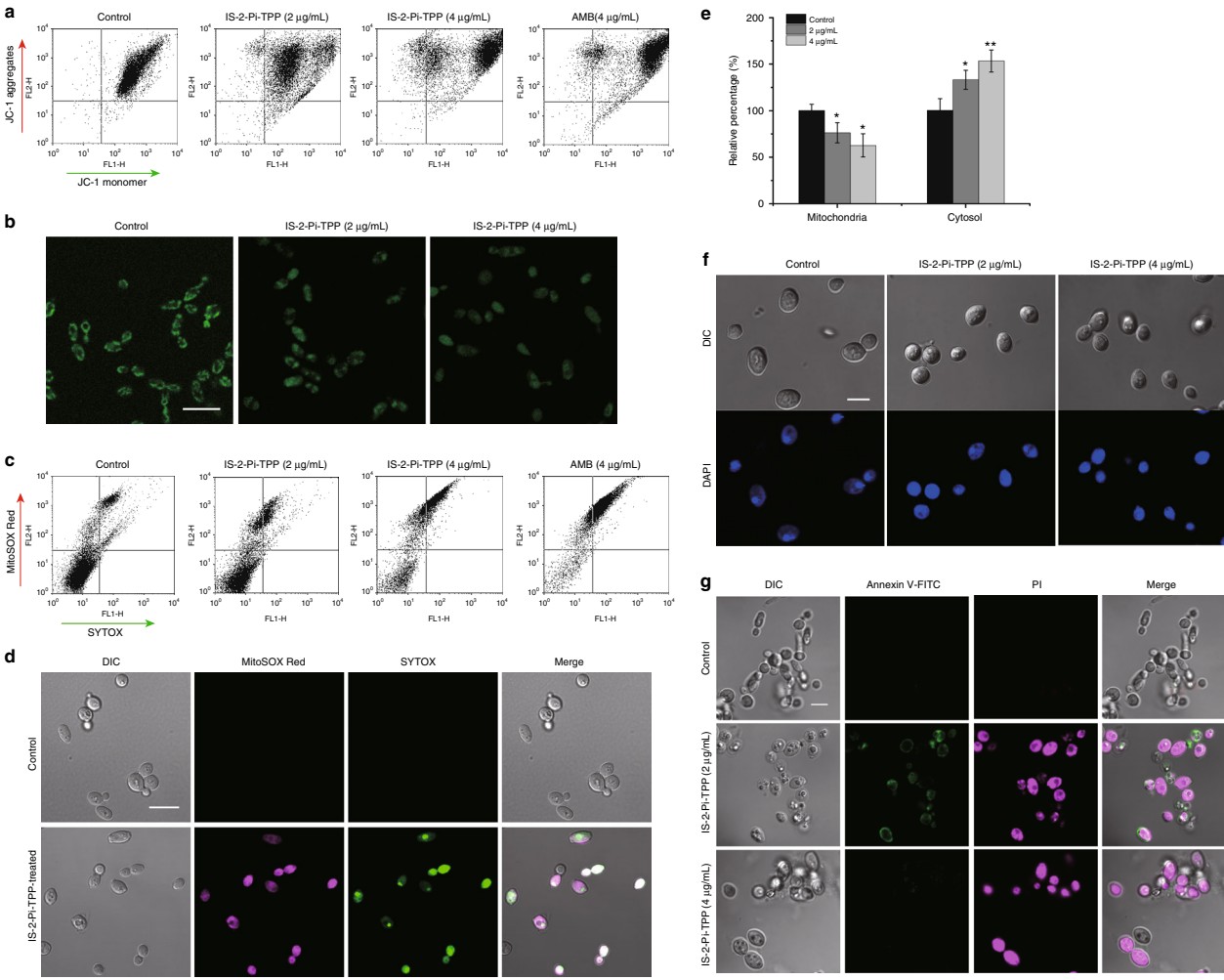

**Fig. 3** The disruptive effect of IS-2-Pi-TPP on mitochondria. **a** The effect of IS-2-Pi-TPP on MMP was monitored by JC-1 staining. JC-1 aggregates accumulate in mitochondrial matrices with high MMP and emit orange-red fluorescence. Upon depolarization of the mitochondrial membrane, JC-1 aggregates are dissipated into the cytoplasm as monomers that emit green fluorescence. **b** The effect of IS-2-Pi-TPP on mitochondrial morphology, as indicated by the Tom70-GFP distribution. Scale bar, 10 μm. **c** and **d** MitoSOX Red and SYTOX were used to indicate the redox state of the mitochondria and the survival state of *C. albicans*, respectively. Superoxide production in the mitochondria and the survival state induced by IS-2-Pi-TPP were detected by flow cytometry (**c**) or observed by confocal microscopy (**d**). Scale bar, 10 μm. **e** The relative contents of Cyt c in the mitochondria and cytosol in cells treated with the indicated concentrations of IS-2-Pi-TPP. Data are shown as means ± s.e.m. from three independent experiments. *$P < 0.05$, **$P < 0.01$ by Student's *t*-test. **f** Nuclear damage of *C. albicans* cells in response to IS-2-Pi-TPP treatment. Scale bar, 5 μm. **g** Apoptosis and necrosis of *C. albicans* induced by IS-2-Pi-TPP. Cells were stained with Annexin V-FITC or PI to detect characteristics of apoptosis or necrosis under treatment with IS-2-Pi-TPP. The stained cells were observed by CLSM. Scale bar, 5 μm. In **a** and **c**, a total of 10,000 events were collected for data analysis without using specific gating strategies

(Fig. 3g). These results suggest that IS-2-Pi-TPP exerts fungicidal activity by inducing both cell apoptosis and necrosis.

**TPP⁺-conjugates have low tendency to induce drug resistance.** To assess the potential for developing drug resistance, serial passage experiments were performed by repeatedly exposing *C. albicans* ATCC10231 to IS-2-Pi-TPP, Sola-TPP, and borneol-TPP. *C. albicans* cells cultured with a sub-lethal concentration (0.5 × MIC) of the tested compounds served as the inoculum for the MIC measurement of the next passage. The MICs of IS-2-Pi-TPP and Sola-TPP remained constant throughout 30 passages. The MIC of borneol-TPP increased by only two-fold after eight passages and remained constant for the subsequent 22 passages (Supplementary Table 8). By contrast, the MIC value of FLC increased from 2 to 64 μg/mL after 30 passages. These results

demonstrate that these TPP⁺-conjugates have a very low tendency to induce drug resistance in *C. albicans*.

**IS-2-Pi-TPP exhibits potent anti-biofilm activity.** Eradicating biofilms of *C. albicans* has attracted particular attention as biofilms are the primary cause of recalcitrant chronic infections[39–41]. We first used the XTT reduction assay to evaluate the ability of IS-2-Pi-TPP to prevent biofilm formation. IS-2-Pi-TPP at 2 μg/mL blocked the formation of *C. albicans* biofilms (Fig. 4a). The Alamar Blue assay was then used to assess the effect of IS-2-Pi-TPP on the cell viability of preformed biofilms. IS-2-Pi-TPP at 16 μg/mL killed almost all cells within the biofilms (Fig. 4b, c). The efficacy of IS-2-Pi-TPP in eradicating *C. albicans* biofilms was confirmed by confocal microscopic observation. Staining with fluorescein diacetate (FDA) and PI demonstrated that the

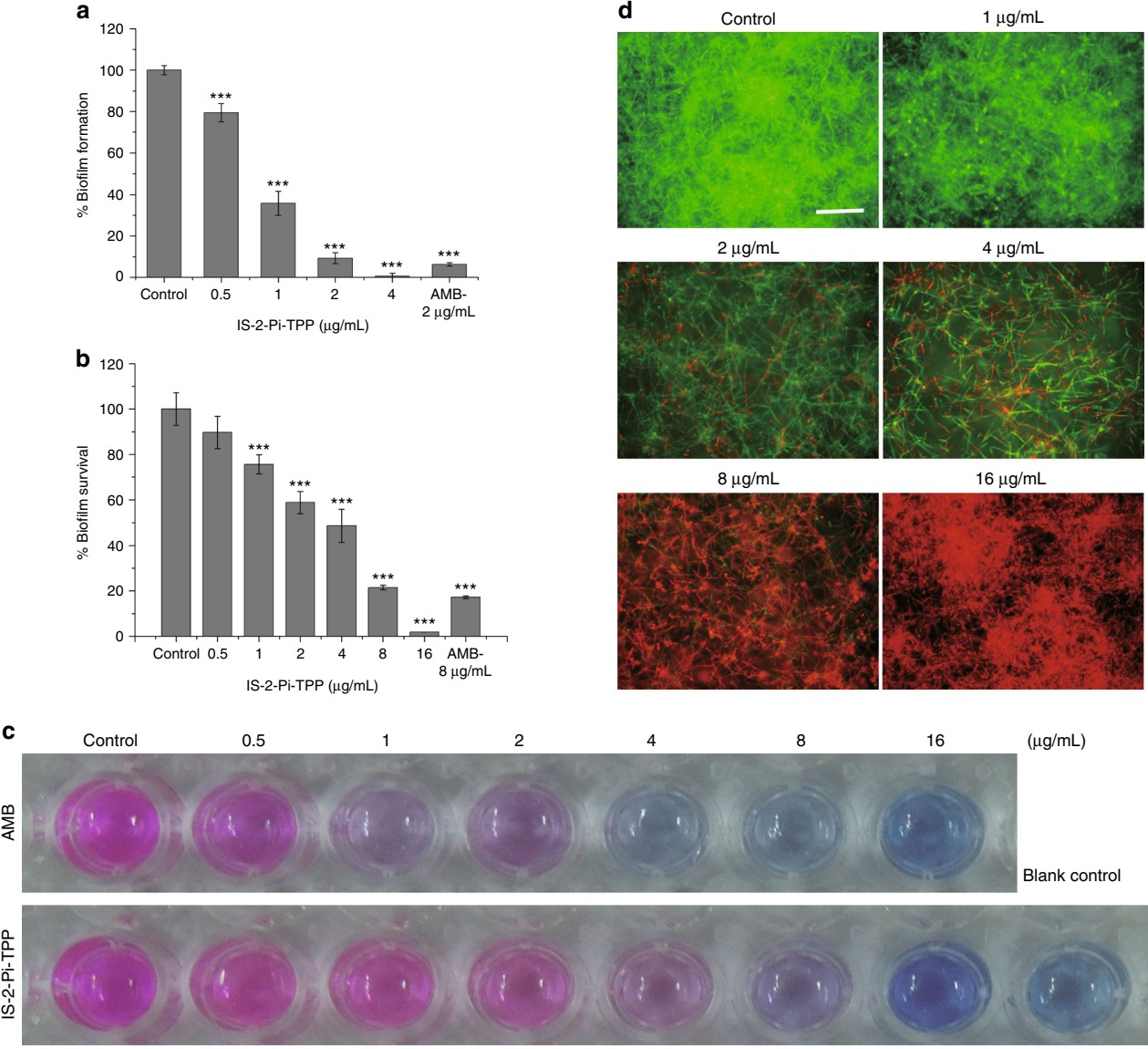

**Fig. 4** The anti-biofilm activity of IS-2-Pi-TPP. **a** The inhibitory effect of IS-2-Pi-TPP on the formation of *C. albicans* biofilms. **b, c** The ability of IS-2-Pi-TPP to eradicate preformed *C. albicans* SC5314 biofilms was tested using the Alamar Blue assay. AMB served as a positive control. **d** IS-2-Pi-TPP-treated *C. albicans* SC5314 biofilms were stained by FDA and PI and subjected to microscopic observation. Scale bar, 100 μm. Data are representative of three independent experiments. In **a** and **b**, data are shown as means ± s.e.m. Student's *t*-test was performed. ***$P < 0.001$ versus the control group

majority of cells within the IS-2-Pi-TPP-treated biofilms were dead (Fig. 4d), in accordance with the results of the biofilm viability assays (Fig. 4b, c). The above results suggest a potential application of IS-2-Pi-TPP in disinfecting preformed *C. albicans* biofilms on biomedical device surfaces.

**The in vivo efficacy of IS-2-Pi-TPP.** We first evaluated the efficacy and toxicity of IS-2-Pi-TPP using *Caenorhabditis elegans* as an infectious model. In the nematode infection assay, IS-2-Pi-TPP at a dose of 1 μg/mL or higher prolonged the survival of *C. albicans*-infected *C. elegans* (Supplementary Figure 15a). Microscopic observation revealed that the worms in the negative control group died and appeared to be rod-shaped, whereas the worms treated with IS-2-Pi-TPP remained in the curly growth state (Supplementary Figure 15b). IS-2-Pi-TPP at a dose of 16 μg/mL or higher resulted in toxicity towards the nematodes (Supplementary Figure 15c). These data suggest a potential

application of IS-2-Pi-TPP in treating *C. albicans* infection in vivo.

Superficial candidiasis manifests as chronic infections of the skin and nails and may result in mucocutaneous candidiasis (CMC). IS-2-Pi-TPP was also evaluated via topical administration in a murine model of skin infections. The advantages of this model over systemic application include an enhanced drug concentration in the infected area and aversion of systemic adverse effects. To enable topical application, we formulated IS-2-Pi-TPP in a 4% (w/w) hypromellose gel. The topical antifungals nystatin (1%) and FLC (1%) were selected as positive controls. IS-2-Pi-TPP (1%) treatment completely eradicated *C. albicans* organisms colonized in murine skin, with efficacy superior to that of FLC (1%) treatment and equal to that of nystatin (1%) treatment (Fig. 5a). The fungal burden results were consistent with histological examinations using periodic acid-Schiff (PAS) staining (Fig. 5b). Histological analysis using hematoxylin and eosin (H&E) staining revealed that there was no obvious inflammatory infiltrates in the skin biopsies of IS-2-Pi-TPP-

treated mice, whereas a high degree of inflammatory infiltrates were observed in the vehicle-treated mice (Fig. 5b). In addition, the PAS staining results revealed that a low dose of IS-2-Pi-TPP (0.25–0.5%) treatment resulted in a filamentation defect of *C. albicans*, implying compromised invasiveness. These results demonstrate that IS-2-Pi-TPP is highly efficacious in a murine model of skin infections.

**The safety of IS-2-Pi-TPP.** Potential adverse effects of the IS-2-Pi-TPP-containing ointment were assessed. Treatment of mice with 8% (w/w) IS-2-Pi-TPP ointment six times within 2 days on shaved intact and abraded skin did not evoke primary irritation. There were no notable changes in behavior and body weight and no signs of systemic toxicity. Histological examination revealed a minimal degree of focal lymphohistiocytic infiltrates in IS-2-Pi-TPP-treated skin (Fig. 5c). Overall, these results indicate that IS-2-Pi-TPP is well-tolerated topically. The efficacy and safety tests suggest that IS-2-Pi-TPP may be useful for treating severe mucosal infections, such as chronic mucocutaneous candidiasis.

## Discussion

A routine antifungal screening bioassay would fail to find PEPSs because compounds that are Cdr substrates are pumped out of cells and cannot be retained intracellularly. In this study, we developed a co-culture assay using strains labeled with two different FPs to enable the characterization of PEPSs whose activity was masked by efflux pumps. Using this assay, two types of

antifungal agents were obtained in one step, and the hit identification rate increased significantly. Chemical strategies that allow PEPSs to bypass expulsion by efflux pumps and restore their activity against wild-type strains are important for antifungal drug development. Glucosylation as a pro-drug is one way to deliver PEPSs into *C. albicans* cells and bypass export by efflux pumps[20]. Here, conjugating PEPSs with TPP[+] restored their antifungal activity against wild-type *C. albicans* and enabled them to overcome antifungal resistance by simultaneously avoiding active efflux and targeting the mitochondria of *Candida* cells.

Drug resistance has become a major public health problem in terms of managing microbial infections[42,43], particularly those resulting from pathogenic fungi due to the limited availability of antifungal drugs[4,44]. This problem necessitates the constant development of new strategies to counteract resistant fungal pathogens. Drug transporters in cell membranes expel drugs or xenobiotics to maintain cellular homeostasis[45]. TPP[+] can function as a carrier to deliver bioactive agents into cells and overcome drug efflux. This characteristic of TPP[+] conjugation does not depend on the targets of the parent compounds (as reviewed by Madak and Neamati)[23]. In this study, we observed that the TPP[+] derivatives passed through the cytoplasmic membrane barrier and accumulated in *C. albicans* cells. The intracellular contents of the TPP[+]-conjugates in wild-type, efflux pump-deficient and efflux pump-hyperactivated strains were maintained at similar levels, although there were some fluctuations in different strains. These results were also consistent with detection

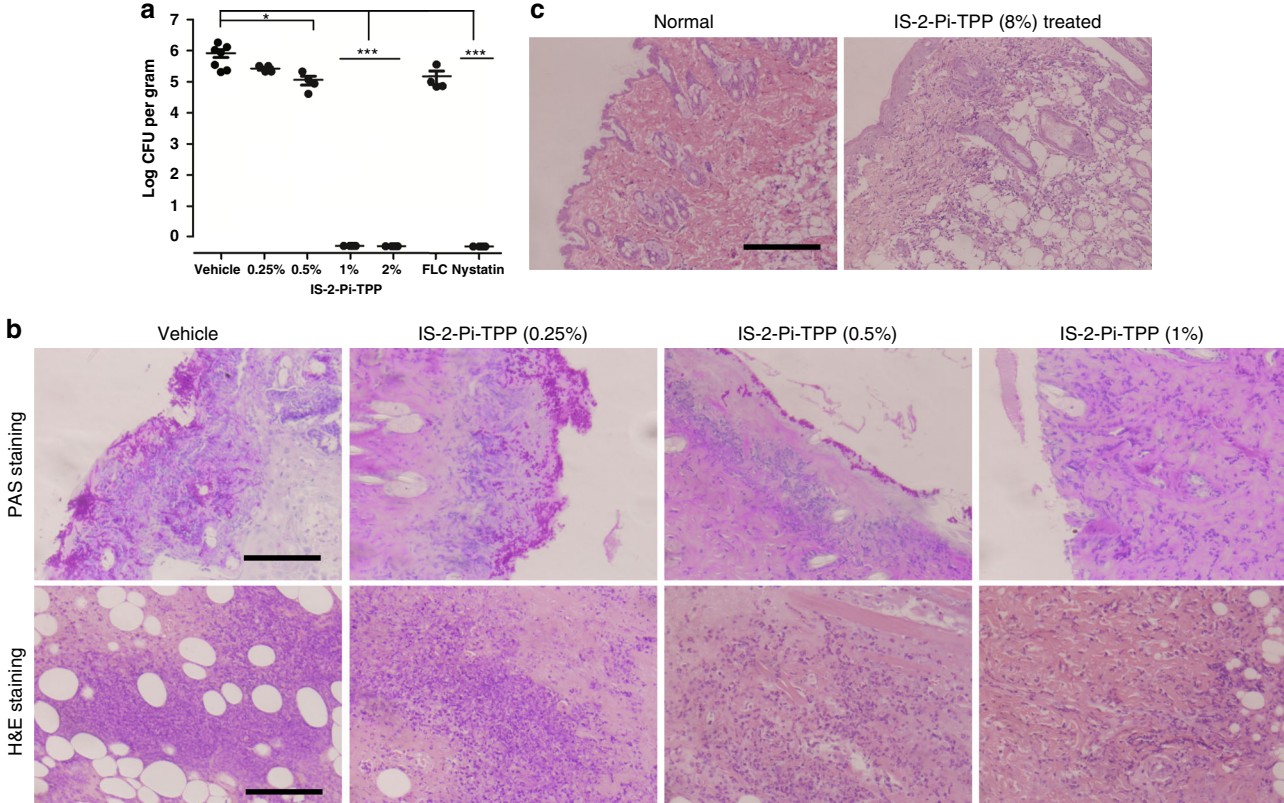

**Fig. 5** Topical application of IS-2-Pi-TPP ointment to treat *C. albicans* murine skin infections. **a** Abraded murine skin was inoculated with 10[7] *C. albicans* SC5314 cells. After 24 h, the skin was treated with ointment containing no drug (vehicle) or 0.25%, 0.5%, 1%, or 2% (w/w) IS-2-Pi-TPP. Ointments containing 1% FLC or nystatin were used as positive controls. The fungal burden in the skin is expressed as the number of viable *C. albicans* cells (log$_{10}$ CFUs per gram) in 4–7 skin samples for each group of mice (seven mice in the vehicle control, four mice in the treated groups). Each circle represents one skin sample. The two-tailed unpaired *t*-test was performed. *$P < 0.05$, ***$P < 0.001$ indicate significance. **b** Micrographs of PAS-stained and H&E-stained skin biopsies after the indicated treatments. Scale bar, 100 μm. **c** Evaluation of the skin toxicity of IS-2-Pi-TPP. Ointment containing 8% IS-2-Pi-TPP was applied on abraded murine skin. H&E staining was used to detect lymphohistiocytic infiltrates under IS-2-Pi-TPP treatment. Scale bar, 100 μm

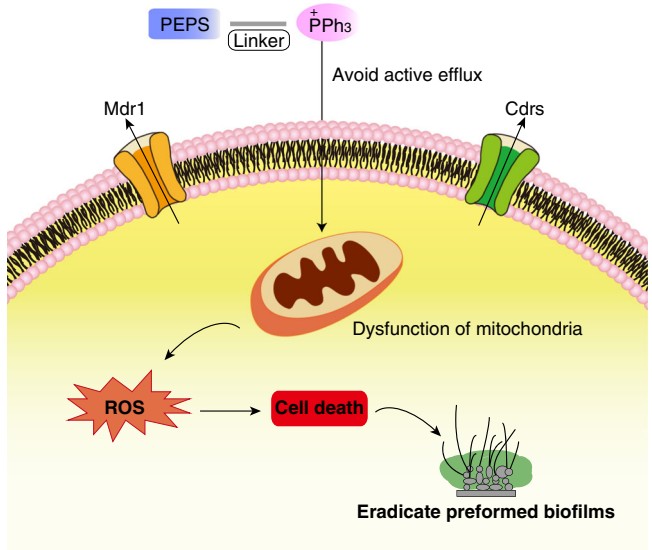

**Fig. 6** The proposed model of antifungal applications of TPP+-conjugates. Cdr substrates with potential antifungal activities are chemically conjugated with TPP+. The TPP+-conjugates regain the ability to kill *C. albicans* wild-type or resistant strains and even can eradicate highly drug-resistant biofilms. The TPP+-conjugates bypass active expulsion by efflux pumps and accumulate in the mitochondria to exert fungicidal activity by inducing mitochondrial dysfunction

using the fluorescent probe RhB-TPP. These lines of evidence, together with the similar levels of antifungal activity of the TPP+-conjugates against different strains, suggest that TPP+-guided transportation is minimally affected by efflux pumps. In addition, the TPP+-conjugates were also active against other types of resistant *C. albicans* strains, supporting a broad application of TPP+-conjugates in addressing drug-resistance problems. Drug-resistance testing showed that *C. albicans* had a low tendency to evolve resistance against the TPP+-conjugates, indicating that TPP+ conjugation may not only address the problem of acquired resistance but also avoid the occurrence of drug resistance in pathogenic fungi.

In addition to their conventional roles in cellular energetics, metabolism, and cell signaling, mitochondria in *C. albicans* are involved in the actions of efflux pumps, oxidative stress responses, iron homeostasis, hyphal growth, and the synthesis of cell membrane components, namely ergosterol and the cell wall[46–48]. Targeting mitochondria is considered an effective approach to combat *Candida* infections, especially those arising from resistant strains[49]. The ability of TPP+ to target the mitochondria supports the promising applications of this moiety in overcoming resistant fungal pathogens. We previously observed that *ent*-kaurane diterpenoids exhibited a dispersed intracellular distribution in tumor cells[50]. The *ent*-beyerene diterpenoids IS-1 and IS-2 share a similar skeleton with oridonin and its analogs, which have been reported to target the mitochondria of cancer cells[51,52]. This similarity suggests that *ent*-beyerene diterpenoids may also localize to the fungal mitochondria and that their antifungal activities are enhanced by increasing their mitochondrial distribution. Therefore, this TPP+ conjugation method may be particularly useful for enhancing or restoring the antifungal activity of compounds targeting the mitochondria. Accumulating studies have shown that many natural products directly target the mitochondrial permeability transition pore complex, respiratory chain, mitochondrial metabolism, or heat-shock protein 90 in

mitochondria (reviewed by Fulda et al.)[53]. Therefore, we believe that TPP+ conjugation will greatly benefit the development of mitochondrial-targeted antifungal drugs from natural products.

Most *Candida* infections are associated with the formation of biofilms, which are highly recalcitrant to antifungal agents, on inert or biological surfaces[54–56]. Biofilms have a complex architecture that provides a safe haven for the persisters within and other opportunistic pathogens and shields them from the immune system[57–59]. This property highlights the need to develop strategies for eradicating biofilms. At a minimum, TPP+-conjugates with the ability to bypass efflux pumps might potentiate azole drugs to eradicate young biofilms, as the cells within young biofilms feature upregulated expression of transporter-encoding genes[60]. In the present study, the TPP+-conjugate IS-2-Pi-TPP alone not only prevented *C. albicans* biofilm formation but also killed preformed mature biofilms. The eradication efficacy of IS-2-Pi-TPP was comparable to that of the clinical drug AMB, suggesting an application in eradicating mature biofilms. In vivo, IS-2-Pi-TPP exhibited potent ability to treat *C. albicans* skin infections comparable to that of the topical antifungal nystatin. The efficacy and safety of IS-2-Pi-TPP support its application in treating some mucosal infections, such as chronic mucocutaneous candidiasis.

In summary, PEPSs, which are often omitted by traditional antifungal screening, can become active against wild-type *Candida* strains upon conjugation with TPP+. The combined use of TPP+ and PEPSs may be a promising approach to develop new antifungal drugs. TPP+ can facilitate the entry of PEPSs into *Candida* cells, a process that is minimally affected by efflux pumps and can be used to address drug-resistance problems (as depicted in Fig. 6). The TPP+-conjugated isosteviol derivative IS-2-Pi-TPP exhibited notably effective activity in killing resistant *Candida* strains, eradicating mature biofilms and treating superficial fungal infections of the skin, indicating its potential application in combating *Candida* infections.

## Methods

**Co-culture assay.** The strains *TDH3-RFP*-CAI4 and *TDH3-GFP*-DSY654 were grown overnight and diluted to $2–5 \times 10^3$ CFU/mL in RPMI1640 medium. The growth curves of the two strains did not differ significantly within a 48-h growth test. The two FP-labeled organisms were then equally mixed and exposed to two-fold serial dilutions of the tested terpene agents ranging from 256 to 2 µg/mL for 48 h at 30 °C. The GFP and RFP fluorescence intensities were excited at 490 and 550 nm, respectively, and their emissions were recorded at 520 and 600 nm, respectively, using a BioTek Synergy H1 microplate reader. The excitation and emission wavelengths for GFP and RFP detection were determined by full wave-length scanning to ensure that the two signals did not interfere with each other. Agents that reduced either the GFP or RFP fluorescence intensity by more than 50% compared with the control group were characterized as hits.

**Measurement of intracellular drug contents.** Exponential-phase SC5314 cells were diluted in RPMI1640 medium to $1 \times 10^8$ cells/mL. After incubation for 20 min, the cells were collected and washed twice with PBS. The cell pellet was dissolved in 250 µL of water and then subjected to three freeze–thaw cycles of 3 min in liquid nitrogen followed by 3 min in a water bath at 65 °C. Methanol (250 µL) was then added to the suspension, which was then disrupted with glass beads in a Precellys 24 homogenizer (Bertin Technologies, Montigny le Bretonneux, France) four times at a speed of $2348 \times g$ (5000 rpm) for 20 s with 10-s intervals. The lysate was stained with Mitotracker Red, a specific marker of mitochondria, to verify that the cells, including the mitochondrial organelles were completely lysed in this method. The lysates was pelleted at $15,871 \times g$ (13,000 rpm) for 2 min at room temperature, and the supernatant was collected. Another 500 µL of methanol was added to the tube for the second round of cell lysis. The supernatant was removed and combined with the previously collected supernatant for freeze drying. The dried sample was then dissolved in 100 µL of methanol and analyzed by HPLC as follows. Chromatographic separation was achieved on a Phenomenex Luna C18 column (250 × 4.6 mm i.d., 5 µm) with a SecurityGuard C18 guard column (4.0 × 3.0 mm i.d., 5 µm; Phenomenex, Torrance, CA) using

mobile phase A, which was composed of 0.5% formic acid (FA) in water, and mobile phase B, which was composed of acetonitrile, at a flow rate of 1.0 mL/min. The HPLC gradient system started with 10% B and linearly increased to 100% B in 30 min, followed by a decrease to 10% B in 2 min prior to column re-equilibration.

**Measurement of intracellular content of Ds-TPP.** Cultures of *C. albicans* cells grown overnight were adjusted to $10^6$ cells/mL with RPMI1640 medium or synthetic medium plus dextrose (SD medium). Ds-TPP or its parent compound dansyl chloride was added to the culture at a final concentration of 50 μg/mL. After 1 h of incubation at 30 °C, the cells were collected, washed twice and viewed under an Olympus fluorescence microscope with a cyan fluorescent protein (CFP) filter set.

**Measurement of intracellular content of RhB-TPP.** *C. albicans* DSY654, SC5314, YEM13, or YEM15 cells grown overnight were adjusted to $10^6$ cells/mL with RPMI1640 medium. RhB-TPP or its parent compound RhB was added to the culture at a final concentration of 20 μg/mL. After 2 h of incubation at 30 °C, the cells were collected, washed twice and analyzed by a FACScan flow cytometer (Becton Dickinson, San Jose, CA) with excitation at 488 nm to measure fluorescence intensity in the FL2 channel as an indicator of intracellular drug content. A total of 10,000 events were collected for data analysis. The resultant data were processed with WinMDI 2.9 software.

**Mitochondrial accumulation test.** Cultures of *C. albicans* cells grown overnight were collected and adjusted to $10^7$ cells/mL with RPMI1640 medium. The tested agents were added at a final concentration of 16 μg/mL, followed by incubation at 30 °C with shaking for 60 min. The cells were then collected by centrifugation and washed twice with PBS. Mitochondria and other intracellular components were isolated from the pellets using a previously described method[61]. Specifically, the cell pellet was suspended in 1 mL of zymolyase buffer containing 50 mM Tris, 10 mM MgCl$_2$, and 1.4 M sorbitol (pH 7.5). DTT was added to a final concentration of 30 mM, and the cells were incubated for 15 min at room temperature. A 5-μL aliquot of zymolyase (5000 U/mL, Zymo Research, USA) was then added to the cell suspension. After incubation for 30 min at 30 °C, the cells were collected by centrifugation at 1000×*g* to obtain yeast protoplasts. The obtained protoplasts were homogenized for 5 min on ice in 0.4 M sorbitol, 0.2% BSA, and 10 mM imidazole with 30–40 strokes of a Dounce homogenizer. The cell lysate was centrifuged at 1000×*g* at 4 °C to remove intact cells. The supernatant was then centrifuged at 12,000×*g* to collect the yeast mitochondria. The supernatant was aspirated into a separate tube to collect other intracellular components. The obtained mitochondria were stained with Mitotracker Red to assess integrity and purity. The drugs distributed in the mitochondria and other intracellular components were analyzed by LC–MS/MS with a SHISEIDO MG III column (150 × 2.1 mm i.d., 5 μm) and a mobile phase of acetonitrile (70%)–water (30%; 5 mM AcNH$_4$, 0.2% FA) under isocratic conditions at a flow rate of 350 μL/min.

**Time-killing kinetics assay.** To explore the fungicidal action of IS-2-Pi-TPP against *C. albicans*, time-killing curves were plotted by measuring the viability of cells treated with IS-2-Pi-TPP or AMB as a positive control. Exponential-phase SC5314 cells were diluted with SD medium to $1 × 10^6$ cells/mL. Serial concentrations of IS-2-Pi-TPP or AMB were added and incubated at 30 °C. The number of viable cells was determined by a colony counting method at specific times. The results are presented as the mean values of triplicate measurements from three independent experiments.

**Drug resistance study.** The drug resistance study was performed by treating *C. albicans* repeatedly with the tested antifungal agents using a previously reported method[62]. Specifically, the MIC values of IS-2-Pi-TPP, Sola-TPP and borneol-TPP against *C. albicans* ATCC10231 were determined with the broth microdilution method described above. *C. albicans* cells taken from duplicate test tubes at a concentration of 0.5 × MIC were adjusted to ~$5 × 10^3$ CFU/mL as the inoculum for the MIC measurement of the next passage. After incubation with the tested antifungal compounds for 24 h at 35 °C, the new MIC values were measured. The process was repeated for 30 passages. All measurements were performed with biological replicates.

**Measurement of MMP.** The MMP in *C. albicans* was assayed by JC-1 staining using a previously reported method[63]. Specifically, *C. albicans* SC5314 cells cultured overnight were treated with different concentrations of IS-2-Pi-TPP ranging from 1 to 8 μg/mL for 2 h. The cells were then washed and suspended in PBS to 5 × $10^6$ cells/mL. Cell suspensions (1 mL) were stained with 10 μg/mL JC-1 (Sigma) at 37 °C for 30 min. The fluorescence densities of the JC-1 aggregates (red) and monomer (green) were recorded by a flow cytometer (FACSCalibur, BD, USA). A total of 10,000 events were collected for data analysis. The resultant data were processed with WinMDI 2.9 software.

**Measurement of ROS.** ROS generation was quantified by MitoSOX Red staining[64]. MitoSOX Red accumulates within the mitochondrial matrix and can be oxidized to a fluorescent product by superoxide. SC5314 cells were diluted to 1 ×

$10^6$ cells/mL with SD medium and exposed to different concentrations of IS-2-Pi-TPP at 30 °C for 3 h. Treatment with 4 μg/mL AMB served as a positive control. After staining with 10 μM MitoSOX Red and 5 μM SYTOX for 30 min in the dark, the cells were collected, and the fluorescence intensity was measured using flow cytometry. A total of 10,000 events were collected for data analysis. The data were processed by WinMDI 2.9 software. In addition, the stained cells were visualized by CLSM (Carl Zeiss, LSM700, Germany) using a ×63 objective lens.

**Effect of IS-2-Pi-TPP on the localization of Tom70-GFP.** To determine the effect of IS-2-Pi-TPP on the localization of Tom70-GFP, CAI4-*TOM70-GFP*[65] was treated with various doses of IS-2-Pi-TPP at 30 °C for 3 h in SD medium. The localization of Tom70-GFP was visualized using CLSM with a ×63 objective lens. An argon laser (488 nm) and band-pass filter (500–560 nm) were used for GFP observation.

**Effect of IS-2-Pi-TPP on Cyt c release from mitochondria.** Cyt c release from mitochondria was measured using a previously reported method[66]. Specifically, *C. albicans* SC5314 cells cultured overnight were treated with different concentrations of IS-2-Pi-TPP ranging from 1 to 8 μg/mL for 4 h. The cells were collected by centrifugation at 5000×*g* for 5 min and washed twice with PBS. The pellet was resuspended in homogenization medium (50 mM Tris (pH 7.5), 2 mM EDTA, and 1 mM phenylmethylsulfonyl fluoride) and homogenized. Then, the homogenate was supplemented with 2% glucose and centrifuged at 2000×*g* for 10 min to remove cell debris and unbroken cells. The supernatant was collected and centrifuged at 30,000×*g* for 45 min. The supernatant was collected to assay Cyt c release from mitochondria to the cytoplasm. The pellet was resuspended in 50 mM Tris (pH 5.0) with 2 mM EDTA, incubated for 5 min, and centrifuged at 5000×*g* for 30 s. The pellet was collected as mitochondria to determine Cyt c remaining in mitochondria. After reduction by ascorbic acid at room temperature for 5 min, the relative quantities of reduced Cyt c in the supernatant and mitochondria were assessed using a spectrophotometer at 550 nm. The protein content was quantified using BSA as a standard.

**Apoptosis and necrosis assay.** Yeast apoptosis and necrosis were detected using a method previously described by our lab[67]. Specifically, cells grown overnight were treated with IS-2-Pi-TPP (0, 2, 4, and 8 μg/mL) for 4 h. The cells were then harvested, washed with sorbitol buffer (1.2 M sorbitol, 0.5 mM MgCl$_2$, 35 mM potassium phosphate, pH 6.8), and digested with 2% glusulase (Sigma, St. Louis, MO, USA) and 15 U/mL lyticase (Sigma, St. Louis, MO, USA) in sorbitol buffer for 2 h at 28 °C. Protoplasts were harvested by centrifugation at 1000×*g* for 10 min, washed in binding buffer (10 mM Hepes/NaOH, pH 7.4, 140 mM NaCl, 2.5 mM CaCl$_2$) containing 1.2 M sorbitol, and resuspended in the same buffer. Then, 2 μL of annexin-FITC and 2 μL of PI (500 mg/mL) were added to 38 μL of cell suspension and incubated for 20 min at room temperature. The cells were harvested, resuspended in binding buffer/sorbitol, applied to a microscop slide, and observed using a Zeiss LSM700 confocal microscope (Carl Zeiss MicroImaging, Thornwood, NY). FITC and PI fluorescence were excited by 488 and 555-nm lasers, respectively. In addition, nuclear damage under IS-2-Pi-TPP treatment was analyzed by DAPI staining[66]. The cells treated with IS-2-Pi-TPP for 4 h were collected, washed twice with PBS buffer, and resuspended in 70% ethanol for brief fixation and permeabilization. After incubation at room temperature for 10 min, the cells were washed with PBS buffer and incubated with 10 μg/mL of DAPI in the dark for 30 min. The stained cells were mounted on a coverslip and observed by CLSM using a ×63 objective.

**Anti-biofilm assay.** To assess the ability of IS-2-Pi-TPP to prevent *C. albicans* biofilms formation, overnight-grown *C. albicans* SC5314 cells were inoculated in 96-well plates at initial densities of $10^6$ cells/mL. After 2 h of adhesion at 37 °C, the adherent cells were aspirated and washed with PBS twice. Fresh RPMI1640 medium containing various doses of IS-2-Pi-TPP was added to the wells for another 48 h of growth at 37 °C. The cells in the different treated groups were quantified using XTT reduction assays[68].

To assess the ability of IS-2-Pi-TPP to eradicate preformed biofilms, *C. albicans* biofilms were allowed to develop in the absence of compound for 48 h, after which non-biofilm material was removed by washing with PBS. Fresh RPMI1640 medium supplemented with the tested agents was added. After an additional 24 h, the biofilms were washed and quantified by the Alamar Blue assay, which is a colorimetric assay involving the cellular reduction of resazurin to resorufin. First, 10% Alamar Blue solution was added to each well. After 2 h of incubation in the dark at 37 °C, images were taken by a camera. The fluorescence was measured with a fluorescence spectrometer at an excitation wavelength of 540 nm and an emission wavelength of 590 nm. The fluorescence values of the samples were corrected by subtracting the average fluorescence value of Alamar Blue in uninoculated wells (blank). The percentage of surviving biofilm cells was calculated relative to the control treatment.

**Analysis of biofilms by fluorescence microscopy.** The *C. albicans* strain SC5314 ($1 × 10^6$ cells/mL) was cultured in RPMI1640 medium for 48 h in the presence of antifungal agents at 37 °C without shaking. The supernatant was aspirated, and

non-adherent cells were removed by thoroughly washing with sterile PBS three times. The *C. albicans* biofilms were then treated with IS-2-Pi-TPP or the same volume of the solvent dimethyl sulfoxide for 24 h. The treated biofilms were stained with FDA and PI for 30 min and washed with PBS three times[69]. The stained samples were observed under an Olympus fluorescence microscope (Olympus IX71, Olympus, Tokyo, Japan). A FITC filter set and a rhodamine filter set were used to excite the green fluorescence of FDA and red fluorescence of PI, respectively. The assessment of cell viability was based on the hydrolysis of FDA by healthy cells, which results in the accumulation of green fluorescence, whereas cells with damaged cell membranes stain fluorescent red due to the intracellular accumulation of PI.

**Murine skin infection model**. This infection model was established using a previously described method with some modifications[70]. Specifically, 6–8-week-old of male Balb/C mice (18–20 g) were used in this study. The animals were maintained and treated under the guidelines approved by the Animal Care and Use Committee of Shandong University. The backs of the mice were shaved using an electric razor and treated with Nair 1 day before the experiment. Fifteen minutes before the experiment, the mice were anesthetized with ketamine and xylazine, and an area of ~2 cm$^2$ in the middle of the back was stripped with sterile Velcro 10 times in succession to produce visibly damaged skin. Five microliters of mid-logarithmic growth-phase *C. albicans* SC5314 containing $1 \times 10^7$ CFUs was applied on the wounded skin. Twenty-four hours after inoculation, ~30 mg of ointment containing 0.25–2% (w/w) IS-2-Pi-TPP was applied on the skin and covered with a circular piece of parafilm. Vehicle was applied on the skin as a negative control and 1% FLC or nystatin was used as a positive control. The ointments were applied three times in a 12-h interval. After a treatment period of 36 h, the mice were euthanized. The skin was separated from the underlying fascia and muscle tissue, and ~2 cm$^2$ of skin from the treated area was excised and divided longitudinally. One half was homogenized in 0.5 mL of saline for quantitative culture. The other half was fixed in zinc-buffered formalin and embedded in paraffin, and thin sections were cut and stained with PAS or H&E for microscopic observations ($n = 7$ in the vehicle control group, $n = 4$ in the treated groups).

**Evaluation of the toxicity of IS-2-Pi-TPP**. Abraded skin was prepared as described above. Ointment containing 8% (w/w) IS-2-Pi-TPP was applied to the abraded skin for six times in 8-h intervals. After treatment, the mice were euthanized, and the treated skin was excised for H&E staining to reveal lymphohistiocytic infiltrates.

**Statistical analysis**. The experimental data were statistically analyzed using Student's *t*-test or the log-rank (Mantel–Cox) test using the SPSS statistical package version 19.0. Statistical significance was set according to the *P*-value: $*P < 0.05$; $**P < 0.01$; and $***P < 0.001$.

Other methods are provided in the SI.

## Data availability

The X-ray crystallographic coordinates for the structures reported in this study have been deposited at the Cambridge Crystallographic Data Centre (CCDC) under deposition number 1813899. The data can be obtained free of charge from the CCDC via www.ccdc.cam.ac.uk/products/csd/request. All other relevant data supporting the findings of this study are available from the corresponding authors on request. A reporting summary for this Article is available as a Supplementary Information file.

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

## Acknowledgements
We thank Professor Joachim Morschhäuser of the University of Würzburg, Professor Kim Lewis of Northeastern University, and Professor Qingguo Qi of Shandong University for donating the *C. albicans* strains used in this study. We thank Professor Judith Berman of the University of Minnesota for providing the plasmids used to construct the fluorescent protein-tagged strains. This work was funded by the National Natural Science Foundation (Nos. 81630093, 81773786, and 81874293) and the Young Scholars Program of Shandong University (2017WLJH41).

## Author contributions
W.C. and H.L. conceived, designed and supervised the study. W.C., M.Z., H.S. and S.Z. constructed the fluorescent protein-labeled strains and performed antifungal tests. W.C. and M.Z. performed flow cytometry analyses and microscopic observations and analyzed the data. M.Z. performed the *C. elegans*–*C. albicans* assay. W.C., J.L. and X.J. conducted mouse experiments. J.L. and A.J. synthesized the TPP-conjugates. J.L. and S.W. collected the spectral data of the synthesized compounds and elucidated their chemical structures. J.L. and Y.G. measured the intracellular drug contents. W.C., J.L. and H.L prepared the figures and co-wrote the paper.

## Additional information

**Competing interests:** The authors declare no competing interests.

