## [Peer Review File · Nature Communications]

Reviewers' comments:

Reviewer #1 (Remarks to the Author):

This manuscript describes the development, and application, of a screen to identify novel antifungals that are substrates of efflux pumps. It goes on to demonstrate how terpenes can be chemically modified with TPP+ to 'bypass' efflux pumps. These modified compounds showed antifungal activity against wild type *Candida albicans* strains and azole-resistant strains overexpressing efflux pumps. The compounds were also effective against a range of *Candida* species. The TPP-conjugated terpene IS-2-Pi-TPP was transported into mitochondria and generated ROS. IS-2-Pi-TPP was effective in preventing *C. albicans* biofilm formation and killed cells in pre-formed biofilms. Finally, IS-2-Pi-TPP showed some activity in overcoming *C. albicans* infections of *Caenorhabditis elegans*.

Fungal infections are a serious clinical problem that is accentuated by the development of drug resistance, as there is a paucity of different classes of antifungal agents. A major antifungal drug resistance mechanism is efflux by ABC or MFS transporters and often resistant isolates overexpress these pumps. One way to circumvent this resistance is to select drugs that are not substrates of efflux pumps. This study takes the opposite approach – to find compounds that are pump substrates and then modify them so that they are no longer transported.

This study describes some well performed experiments that identify and optimize promising antifungals, but I do not feel that it demonstrates important advances of general interest.

Major points

1. This manuscript has some novel approaches, such as looking for antifungals that are substrates of efflux pumps and the targeting of compounds to mitochondria in order to prevent their efflux, but I am not convinced that the combination of these approaches will be of wide benefit. If the antifungals are substrates of efflux pumps this is a potential disadvantage as efflux is a common resistance mechanism that will need to be overcome. Directing these drugs to mitochondria will not work if the targets of these novel antifungals are elsewhere in the cell, as is likely to be the case.
2. The methods in the main manuscript are very brief with additional methods and data in supplementary material. This supplementary information is very extensive running to 87 pages. This makes it quite hard to locate some of the information. For instance, I found it hard to work out the resistance genotype of strains YEM13 and YEM15 (line 102). It would help if this information was included in the footnotes to the Tables in the main manuscript.
3. In lines 123 – 132 the authors report the intracellular accumulation of TPP+ -conjugates. Do they mean cytoplasmic accumulation or uptake into mitochondria? When they freeze/thaw the cells, treat them with methanol and break them with glass beads are they likely to release the conjugate from mitochondria? The authors should discuss this possibility – the extracts could be checked for the presence of mitochondrial markers. The authors later separate the mitochondria and measure the amount of conjugates in extracts, but provide no evidence that they obtain pure mitochondrial preparations.
4. There is no discussion of the possible toxicity of TPP-conjugates to human cells. TPP is used to target drugs to cancer cells. It would seem possible that the TPP-terpenes could be targeted to human mitochondria and have a similar effect on ROS. I would like to see some toxicity data. I appreciate that the conjugates were used in a *C. elegans* infection assay, but there was not a great improvement in survival at the concentrations of conjugate used and perhaps at higher concentrations the conjugate would have been toxic to the *C. elegans* cells.

Minor points

1. I am not convinced that 'bypassing efflux pumps' is the best phrase to use in this manuscript. These efflux pumps pump compounds out of the cell. To my understanding, if compounds bypass

the pumps they exit the cell by another mechanism. This is not what the authors mean. Perhaps 'avoiding efflux' would be a better term.

2. It is widely known that drug resistant *C. albicans* cells often over-express efflux pumps, however most strains express Cdr1 and Cdr2 to some extent. That is why the authors compare wild-type strains (as opposed to resistant strains) to pump-deleted cells. This might confuse some readers; I think the authors should point out that most strains express pumps at a low level constitutively.

3. Lines 54-55, this sentence is clumsy. Perhaps it should read "However, there are some disadvantages from chemical modification of antifungals such as lower selectivity and reduced antifungal activity".

4. Line 58, replace "achieved" with 'identified'.

5. Lines 131-132, it appears to me that the intracellular conjugate contents of YEM13 and YEM15 actually vary quite a bit from wild-type cells, for example 370% for IS-2-TPP and YEM13 (Supplementary Table 4).

6. Line 170, IS-2-Pi-TPP is not referred to as compound 35 on Supplementary Table 6.

Reviewer #2 (Remarks to the Author):

The article by J Liu et al presents an attractive and novel strategy to develop antifungals which can bypass efflux pumps. Overall the study holds merit and has sufficient implications in antifungal therapy. However, in order to draw conclusions, certain points need to be addressed prior to considering the article for publication.

1. The potential efflux pump substrates (PEPS) need to be addressed in detail. The authors should employ some additional methods to propose the efflux pumps (Cdr1, Cdr2, Mdr1 or all three) which interact with the base molecules (and avoid interaction with TPP conjugates).

2. Assays should be employed to assess the tendency of resistance development against the selected most potent molecules.

3. Effect on preformed biofilms should be investigated with Scanning electron microscopy to provide better insights.

4. In vivo efficacy should be tested in a mouse model to judge their potential in antifungal therapy. At least cytotoxicity assay should definitely be performed against human cell lines if mouse model is not possible.

Reviewer #3 (Remarks to the Author):

J. Liu, W. Cheng, H. Lou and co-workers report their approach to: (1) mining natural products with antifungal activity using a mixed assay of both wild-type and Cdr-deficient strains of *C. albicans*; and (2) overcoming the effect of efflux pumps, and efflux pump upregulation, as a mechanism of drug resistance in *C. albicans* by ligation of a phosphonium moiety to terpene substrates with antifungal activity. The authors present results that clearly demonstrate these two outcomes, and the quality and breadth of experimental data, especially the synthetic experiments undertaken (my field of expertise), is of sufficient standard.

While the authors do a very good job of summarising and citing literature on efflux pump associated mechanisms of antifungal resistance, there is insufficient detail of prior literature related to mixed assay screening and phosphonium-tethered approaches to enhancing drug performance in the introductory section. Unfortunately, these omissions make it very difficult for the reader to develop a contextual understanding and overview of the field before reading more about the authors' current contribution. It also makes it challenging for the reader to gain an appreciation for the true significance and novelty of this work.

It is of utmost importance that the introductory section empowers the reader with an understanding of previous work in the field; and clearly articulates the significance of the current study in the context of prior literature. This is especially important when publishing work in a journal such as Nature Communications, which has a very broad readership.

For this reason, I must recommend that the authors consider my comments below and resubmit the manuscript with major revisions to its current form.

(Comment 1)

The text throughout the manuscript has been carefully edited, with consistent style and presentation. However, there are some instances where the use of language is unintentionally misleading or incomplete. The manuscript should be edited to correct for all such instances.

A few examples include:

Line 127 "The intracellular contents of TPP+ -conjugates accumulated a lot in..."

This sentence can be confusing to read and lacks some information. The TPP+ -conjugates themselves do not have an intracellular content, which is the literal message given at the beginning of this sentence.

Line 83 "...due to the structural and biological diversity of terpenes".

In this sentence: what is due to the structural and biological diversity of terpenes? I can only think that the intention of this sentence may have been "...as a collective representation of the structural and biological diversity of terpenes".

(Comment 2)

Further to comment (1), there are some phrases used in the text that are vague or ambiguous and should be written with more information, or greater clarity, so that the reader is better informed. For example, the first sentence of the abstract reads "Drug-resistance resulting from efflux pumps is prevalent in clinical *Candida* isolates, highlighting the efforts of developing novel antifungal agents". It is of course acceptable that the first sentence of the abstract is general, but it must be informative and clear. This sentence should inform the reader more clearly that it is the expression (or upregulation) of efflux pumps in clinical *Candida* isolates that has led to the development of drug resistance and that there is a need for new antifungal agents to overcome this.

There are in fact many instances of vague or ambiguous passages in the abstract. Another example is in line 20: "...will broaden the way to find more antifungal agents". In this case, the word "way" is simply too ambiguous for any reader to understand the authors' intention. Similar instances throughout the manuscript should be addressed.

(Comment 3)

As mentioned earlier, the introductory section of the manuscript omits a detailed overview of literature on: (1) mixed assay screening for antimicrobial compounds; and (2) phosphonium-based approaches to drug delivery. As a result, the current work is presented without proper context, and the reader has little understanding of its true relevance or significance. Perhaps the most confusing aspect for me was to read line 60, where TPP+ is first mentioned, followed by the comment "to our surprise". There is no prior context to phosphonium-based approaches in the text and this comment led me to believe that the authors have made a unique discovery. Later, within the results section (line 95–98), I found that there is literature on phosphonium-based methods for therapeutic application, and that the authors' motivation to use phosphine appeared to be well founded (and was indeed by design). Since the authors' comments here are only brief, I am then led to ask the question: exactly how novel or significant are the results presented here? The reader should not feel this way when reading a Nature Communications article.

This study needs to be clearly placed in the context of prior literature and its significance must be made obvious to the reader. The overall contextual argument needs to be outlined in the introductory section.

(Comment 4)

In this manuscript, it is claimed that the antifungal activity of PEPS is “restored” when tethered to a triphenylphosphonium moiety. But, by which mechanism/s are the PEPS-TPP+ conjugates fungicidal? The word “restored” may suggest that the TPP+ moiety is itself benign, and only acts to localise the terpene ‘warhead’ in the mitochondria. While this may indeed be the case, there is little evidence presented here to convince the reader.

It is demonstrated that methyl triphenylphosphonium bromide has no antifungal activity; however, to the best of my knowledge, this compound is a profoundly poor choice for a negative control. It is my understanding that the physicochemical properties of MePPh₃Br are very different from the PEPS-TPP+ conjugates reported here. It would be far more convincing to test a range of terpene-TPP+ conjugates that were inactive in the Cdr-deficient strains from the primary screen, yet have a somewhat similar structure to the active PEPS. If it were in fact the case that TPP+ ligation has no effect on activity, one would expect a null result. However, if the large terpene(non-PEPS)-TPP+ conjugates are fungicidal through a different mechanism, conjugates with an ‘inactive warhead’ may still illicit the desired effect.

This point of contention is critically relevant to the argument presented in this manuscript, and must be addressed. It may be useful to conduct additional experiments and obtain a better understanding of the mechanisms of fungicidal activity (even at a simplistic level). But, regardless of whether or not more results are obtained, a clear in-text explanation of any important conceptual or experimental limitation is necessary.

(Comment 5)

Finally, some of the images (especially figures 1 and 2) are small and poorly resolved such that they are very difficult to read. This may have been caused through generating the pdf for review, but it may also be an issue for publication. These images should be checked for appropriate size and quality.

Reviewers' comments:

Reviewer #1 (Remarks to the Author):

Major points

1. This manuscript has some novel approaches, such as looking for antifungals that are substrates of efflux pumps and the targeting of compounds to mitochondria in order to prevent their efflux, but I am not convinced that the combination of these approaches will be of wide benefit. If the antifungals are substrates of efflux pumps this is a potential disadvantage as efflux is a common resistance mechanism that will need to be overcome. Directing these drugs to mitochondria will not work if the targets of these novel antifungals are elsewhere in the cell, as is likely to be the case.

Answer: In this study, we found that some potential substrates of efflux pumps lost their antifungal activity due to active export by efflux pumps. TPP⁺ conjugation enabled these efflux pump substrates to avoid efflux and restored their antifungal activity against both wild-type and efflux pump-hyperactivated strains. These results suggest that TPP⁺ conjugation is an alternative strategy for overcoming antifungal resistance arising from efflux pumps.

The diterpenoids used in this study share similar skeletons with oridonin, which has been reported to target the mitochondria of cancer cells (references 1 and 2). TPP⁺ conjugation not only increased the accumulation of these diterpenoids in *Candida* cells by avoiding efflux but also enhanced their activity by increasing the mitochondrial distribution. In addition, several classes of natural products directly target the mitochondrial permeability transition pore complex, respiratory chain, mitochondrial metabolism or heat-shock protein 90 in the mitochondria (references 3-10). The bioactivities of these mitochondrial-targeted compounds, such as metformin (reference 11), can be greatly increased by TPP⁺ conjugation by increasing their accumulation in mitochondria and reducing off-target effects.

For those agents with targets other than the mitochondria, TPP⁺ can function as a carrier to deliver them into cells and increase intracellular concentrations by avoiding efflux. Many cases have been reported in antitumor studies (References 12-17). For example, doxorubicin targets topoisomerase II in the nucleus of tumor cells and is susceptible to efflux pumps. Addition of the

TPP⁺ group improved the antitumor activity of doxorubicin by increasing intracellular accumulation (Reference 12). Therefore, we believe that TPP⁺ conjugation will greatly benefit the development of antifungal drugs. We also added relevant discussions in the discussion section in revised manuscript.

References:

1. Xu, S., Luo, S., Yao, H., Cai, H., Miao, X., & Wu, F., et al. (2016). Probing the anticancer action of oridonin with fluorescent analogues: visualizing subcellular localization to mitochondria. *Journal of Medicinal Chemistry*, 59(10), 5022.
2. Xu, S., Yao, H., Luo, S., Zhang, Y. K., Yang, D. H., & Li, D., et al. (2017). A novel potent anticancer compound optimized from a natural oridonin scaffold induces apoptosis and cell cycle arrest through the mitochondrial pathway. *Journal of Medicinal Chemistry*, 60(4), 1449.
3. Fulda, S., Galluzzi, L., & Kroemer, G. (2010). Targeting mitochondria for cancer therapy. *nat rev drug discov* 9:447-464. *Nature Reviews Drug Discovery*, 9(6), 447-464.
4. Di, S. R. (2010). Natural products as antifungal agents against clinically relevant pathogens. *Cheminform*, 41(43), 1084.
5. Gorlach, S., Fichna, J., & Lewandowska, U. (2015). Polyphenols as mitochondria-targeted anticancer drugs. *Cancer Letters*, 366(2), 141.
6. Beraud, E., & Chandy, K. G. (2011). Therapeutic potential of peptide toxins that target ion channels. *Inflammation & Allergy Drug Targets*, 10(5), 322-42.
7. Leanza, L., Romio, M., Becker, K. A., Azzolini, M., Trentin, L., & Managò, A., et al. (2017). Direct pharmacological targeting of a mitochondrial ion channel selectively kills tumor cells in vivo. *Cancer Cell*, 31(4), 516.
8. McLellan, C. A., Vincent, B. M., Solis, N. V., Lancaster, A. K., Sullivan, L. B., & Hartland, C. L., et al. (2017). Inhibiting mitochondrial phosphate transport as an unexploited antifungal strategy. *Nature Chemical Biology*, 14(2).
9. Yeaman, M. R., Büttner, S., & Thevissen, K. (2018). Regulated cell death as a therapeutic target for novel antifungal peptides and biologics. *Oxidative Medicine & Cellular Longevity*, 2018, 1-20.
10. Ye, Y., Zhang, T., Yuan, H., Li, D., Lou, H. X., & Fan, P. (2017). Mitochondria-targeted lupane triterpenoid derivatives and their selective apoptosis-inducing anticancer mechanisms. *Journal of Medicinal Chemistry*, 60(14), 6353.

11. Boukalova, S., Stursa, J., Werner, L., Ezrova, Z., Cerny, J., & Bezaworkgeleta, A., et al. (2016). Mitochondrial targeting of metformin enhances its activity against pancreatic cancer. *Molecular Cancer Therapeutics*, 15(12), 2875-2886.
12. Han, M.; Vakili, M. R.; Soleymani Abyaneh, H.; Molavi, O.; Lai, R.; Lavasanifar, A. (2014). Mitochondrial Delivery of Doxorubicin via Triphenylphosphine Modification for Overcoming Drug Resistance in MDA-MB-435/DOX Cells. *Mol. Pharm.*, 11, 2640-2649.
13. Chamberlain, G. R.; Tulumello, D. V.; Kelley, S. O. (2013). Targeted Delivery of Doxorubicin to Mitochondria. *ACS Chem. Biol.*, 8, 1389-1395.
14. Walle, T.; Vincent, T. S.; Walle, U. K. (2003). Evidence of Covalent Binding of the Dietary Flavonoid Quercetin to DNA and Protein in Human Intestinal and Hepatic Cells. *Biochem. Pharmacol.* 65, 1603-1610.
15. Reddy, C. A.; Somepalli, V.; Golakoti, T.; Kanugula, A. K.; Karnewar, S. et al. (2014). Mitochondrial-Targeted Curcuminoids: A Strategy to Enhance Bioavailability and Anticancer Efficacy of Curcumin. *PLoS One*, 9, e89351.
16. Mattarei, A.; Biasutto, L.; Marotta, E.; De Marchi, U.; Sassi, N. et al. (2008). A Mitochondriotropic Derivative of Quercetin: A Strategy to Increase the Effectiveness of Polyphenols. *Chembiochem Eur. J. Chem. Biol.* 9, 2633-2642.
17. Jara, J. A.; Castro-Castillo, V.; Saavedra-Olavarría, J.; Peredo, L.; Pavanni, M. et al. (2014). Kemmerling, U.; Maya, J. D.; Ferreira, J. Antiproliferative and Uncoupling Effects of Delocalized, Lipophilic, Cationic Gallic Acid Derivatives on Cancer Cell Lines. Validation *in vivo* in Singenic Mice. *J. Med. Chem.* **2014**, 57, 2440-2454.

2. The methods in the main manuscript are very brief with additional methods and data in supplementary material. This supplementary information is very extensive running to 87 pages. This makes it quite hard to locate some of the information. For instance, I found it hard to work out the resistance genotype of strains YEM13 and YEM15 (line 102). It would help if this information was included in the footnotes to the Tables in the main manuscript.

Answer: Thank you for your suggestions. We have added the detailed information on the strains in the footnotes to the Tables.

3. In lines 123 – 132 the authors report the intracellular accumulation of TPP⁺-conjugates. Do they mean cytoplasmic accumulation or uptake into mitochondria? When they freeze/thaw the cells, treat them with methanol and break them with glass beads are they likely to release the conjugate from mitochondria? The authors should discuss this possibility – the extracts could be checked for the presence of mitochondrial markers. The authors later separate the mitochondria and measure the amount of conjugates in extracts, but provide no evidence that they obtain pure mitochondrial preparations.

Answer: In lines 123-132, the intracellular accumulation of TPP⁺-conjugates refers to the total drug distribution in the mitochondria and other cellular compartments. We have clarified this point in the revised manuscript. To obtain the total intracellular contents of the tested agents, we utilized the glass bead disruption method. To ensure that the cells, including the mitochondrial organelles, were completely lysed using this method, the cell lysates were stained with MitoTracker Red, a specific marker of mitochondria. Microscopic examination did not reveal any stained cells or organelles, indicating that the cells were completely lysed.

We utilized zymolyase to remove the cell wall and extract the mitochondria. Under this treatment, we observed a great quantity of MitoTracker Red-stained organelles, indicating that the mitochondria remained intact. We have added our verification methods in the Methods section.

4. There is no discussion of the possible toxicity of TPP-conjugates to human cells. TPP is used to target drugs to cancer cells. It would seem possible that the TPP-terpenes could be targeted to human mitochondria and have a similar effect on ROS. I would like to see some toxicity data. I appreciate that the conjugates were used in a *C. elegans* infection assay, but there was not a great improvement in survival at the concentrations of conjugate used and perhaps at higher concentrations the conjugate would have been toxic to the *C. elegans* cells.

Answer: We performed *in vivo* tests of the TPP⁺-conjugate IS-2-Pi-TPP using murine skin and systemic infection models. IS-2-Pi-TPP exhibited potent ability to treat *C. albicans* skin infections comparable to that of the topical antifungal nystatin. In addition, IS-2-Pi-TPP did not evoke primary irritation or significant focal lymphohistiocytic infiltrates in the dermis of the wounded animals, suggesting that IS-2-Pi-TPP is well-tolerated topically. These results may be meaningful for treating severe mucosal infections such as chronic mucocutaneous candidiasis. In a systemic

infection model, IS-2-Pi-TPP did not significantly improve the survival of *C. albicans*-infected animals, possibly due to its poor pharmacokinetic profile, indicating that IS-2-Pi-TPP is not suitable for treating disseminated candidiasis.

Minor points

1. I am not convinced that ‘bypassing efflux pumps’ is the best phrase to use in this manuscript. These efflux pumps pump compounds out of the cell. To my understanding, if compounds bypass the pumps they exit the cell by another mechanism. This is not what the authors mean. Perhaps ‘avoiding efflux’ would be a better term.

Answer: In the revised manuscript, we have changed this phrase as suggested.

2. It is widely known that drug resistant *C. albicans* cells often over-express efflux pumps, however most strains express Cdr1 and Cdr2 to some extent. That is why the authors compare wild-type strains (as opposed to resistant strains) to pump-deleted cells. This might confuse some readers; I think the authors should point out that most strains express pumps at a low level constitutively.

Answer: In accordance with your suggestions, we have explained that the wild-type strain SC5314 expresses pump-encoding genes at a low level under non-inducing conditions in the revised manuscript.

3. Lines 54-55, this sentence is clumsy. Perhaps it should read “However, there are some disadvantages from chemical modification of antifungals such as lower selectivity and reduced antifungal activity”.

Answer: In the revised manuscript, we have changed this sentence to "However, these strategies are associated with some disadvantages, such as lower selectivity and reduced antifungal activity."

4. Line 58, replace “achieved” with ‘identified’.

Answer: We have made the requested change in the revised manuscript.

5. Lines 131-132, it appears to me that the intracellular conjugate contents of YEM13 and YEM15 actually vary quite a bit from wild-type cells, for example 370% for IS-2-TPP and YEM13 (Supplementary Table 4).

Answer: The permeability driving forces for TPP⁺ analogues are the cell membrane potential and

mitochondrial membrane potential, which provide a sufficient force to overcome drug efflux mechanisms (reference 1). In this study, we found that the intracellular contents of the TPP⁺-conjugates were minimally affected by efflux pumps, and this feature can be used to overcome resistance mechanisms. However, we did not rule out the possibility that the TPP analogues can be exported by efflux pumps to a lesser extent. Generally, the intracellular contents of tested TPP⁺-conjugates in wild-type, efflux pump-deficient and efflux pump-hyperactivated strains were maintained at similar levels, although there were some fluctuations in different strains. The discrepancies in the intracellular accumulation of IS-2-TPP among the different strains may be attributed to its chemical structure.

References

1. Madak, J. T., & Neamati, N. (2015). Membrane permeable lipophilic cations as mitochondrial directing groups. *Current Topics in Medicinal Chemistry*, 15(8), 745-766.

6. Line 170, IS-2-Pi-TPP is not referred to as compound 35 on Supplementary Table 6.

Answer: We have added the activity of compound **35** in Supplementary Table 6 in the revised manuscript.

Reviewer #2 (Remarks to the Author):

1. The potential efflux pump substrates (PEPS) need to be addressed in detail. The authors should employ some additional methods to propose the efflux pumps (Cdr1, Cdr2, Mdr1 or all three) which interact with the base molecules (and avoid interaction with TPP conjugates).

Answer: We failed to provide direct evidence that the efflux pumps interact with the parent molecules but not the TPP⁺-conjugates. However, several lines of indirect evidence, including new data, demonstrate that the PEPSs are exported by efflux pumps and that the exit of TPP⁺-conjugates is minimally affected by efflux pumps.

First, the PEPSs in this study were only active against efflux pump-deficient strains, whereas their TPP⁺-conjugates were active against wild-type strains, efflux pump-activated strains and

efflux pump-deficient strains. PEPSs traverse the *C. albicans* membrane into cells, probably through passive diffusion or other routes. However, once inside the cell, PEPSs are susceptible to efflux pumps, and the diffusion force cannot overcome export by efflux pumps. By contrast, the permeability driving forces for TPP⁺-conjugates are the cell membrane potential and mitochondrial membrane potential. The membrane potential provides sufficient force to overcome drug efflux mechanisms in tumor cells (as reviewed in reference 1). For example, the addition to doxorubicin of a mitochondrial-directing group, either TPP⁺ or a mitochondrial-targeted peptide, resulted in improved intracellular accumulation in tumor cells, including cells overexpressing the p-glycoprotein efflux pump (references 2 and 3).

Second, the intracellular contents of the TPP⁺-conjugates were similar among the wild-type strain, efflux pump-activated strains and efflux pump-deficient strains, suggesting that the TPP⁺-conjugates are minimally affected by efflux pumps.

Third, a (dansyl)-labeled TPP⁺-conjugate accumulated in *C. albicans* cells, whereas dansyl chloride was excluded.

In addition, we synthesized a new fluorescent probe, RhB-TPP, in which the TPP⁺ group was conjugated to rhodamine B, a fluorescent substrate of multidrug transporters (reference 4). We then utilized flow cytometry to analyze the intracellular contents of RhB and TPP-RhB in wild-type, efflux pump-hyperactivated and efflux pump-deficient strains. The intracellular contents of RhB were higher in the efflux pump mutant strain than in the other strains, indicating RhB is susceptible to efflux pumps. Compared with RhB, the intracellular contents of RhB-TPP were notably higher and were maintained at similar levels in the different strains, suggesting that the export of TPP⁺-conjugates is minimally affected by efflux pumps. We have added the new data in Supplementary Fig. 4 and added relevant discussions in the discussion section.

References

1. Madak, J. T., & Neamati, N. (2015). Membrane permeable lipophilic cations as mitochondrial directing groups. *Current Topics in Medicinal Chemistry*, 15(8), 745-766.
2. Han, M.; Vakili, M. R.; Soleymani Abyaneh, H.; Molavi, O.; Lai, R.; Lavasanifar, A. (2014). Mitochondrial Delivery of Doxorubicin via Triphenylphosphine Modification for Overcoming Drug Resistance in MDA-MB-435/DOX Cells. *Mol. Pharm.*, 11, 2640-2649.

3. Chamberlain, G. R.; Tulumello, D. V.; Kelley, S. O. (2013). Targeted Delivery of Doxorubicin to Mitochondria. *ACS Chem. Biol.*, 8, 1389-1395.

4. Luckenbach, T., Corsi, I., & Epel, D. (2004). Fatal attraction: synthetic musk fragrances compromise multixenobiotic defense systems in mussels. *Marine Environmental Research*, 58, 215-219.

2. Assays should be employed to assess the tendency of resistance development against the selected most potent molecules.

Answer: The resistance development test has been performed according to your suggestion. *C. albicans* did not evolve drug resistance even after 30 passages under treatment with the TPP⁺-conjugates IS-2-Pi-TPP and Sola-TPP. The MIC value of borneol-TPP against *C. albicans* increased only twofold during 30 passages of drug-resistance induction. These results demonstrate that these TPP⁺-conjugates have a very low tendency to induce drug resistance in *C. albicans*.

3. Effect on preformed biofilms should be investigated with Scanning electron microscopy to provide better insights.

Answer: We performed scanning electron microscopy to inspect the preformed biofilms exposed to IS-2-Pi-TPP. However, we did not observe any morphological alterations of the treated biofilms compared with vehicle-treated biofilms. This lack of alteration is likely due to killing of the cells within the biofilms by IS-2-Pi-TPP rather than disruption of the biofilm structure.

4. In vivo efficacy should be tested in a mouse model to judge their potential in antifungal therapy. At least cytotoxicity assay should definitely be performed against human cell lines if mouse model is not possible.

Answer: We performed *in vivo* tests of the TPP⁺-conjugate IS-2-Pi-TPP using murine skin and systemic infection models. IS-2-Pi-TPP exhibited potent ability to treat *C. albicans* skin infections comparable to that of the topical antifungal nystatin. In addition, IS-2-Pi-TPP did not evoke primary irritation or significant focal lymphohistiocytic infiltrates in the dermis of the wounded animals, suggesting that IS-2-Pi-TPP is well-tolerated topically. These results may be meaningful

for treating severe mucosal infections such as chronic mucocutaneous candidiasis. In the systemic infection model, IS-2-Pi-TPP did not significantly improve the survival of *C. albicans*-infected animals, indicating that IS-2-Pi-TPP is not suitable for treating disseminated candidiasis.

Reviewer #3 (Remarks to the Author):

While the authors do a very good job of summarising and citing literature on efflux pump associated mechanisms of antifungal resistance, there is insufficient detail of prior literature related to mixed assay screening and phosphonium-tethered approaches to enhancing drug performance in the introductory section. Unfortunately, these omissions make it very difficult for the reader to develop a contextual understanding and overview of the field before reading more about the authors' current contribution. It also makes it challenging for the reader to gain an appreciation for the true significance and novelty of this work.

It is of utmost importance that the introductory section empowers the reader with an understanding of previous work in the field; and clearly articulates the significance of the current study in the context of prior literature. This is especially important when publishing work in a journal such as Nature Communications, which has a very broad readership.

Answer: We have added information on the mixed assay screening and phosphonium-tethered approaches in the introduction section in the revised version.

For this reason, I must recommend that the authors consider my comments below and resubmit the manuscript with major revisions to its current form.

(Comment 1)

The text throughout the manuscript has been carefully edited, with consistent style and presentation. However, there are some instances where the use of language is unintentionally misleading or incomplete. The manuscript should be edited to correct for all such instances.

Answer: We have asked a native English speaker Dr. Dawn Schmidt to critically read our manuscript and correct inappropriate language.

A few examples include:

Line 127 “The intracellular contents of TPP+-conjugates accumulated a lot in...”

This sentence can be confusing to read and lacks some information. The TPP+-conjugates themselves do not have an intracellular content, which is the literal message given at the beginning of this sentence.

Line 83 “...due to the structural and biological diversity of terpenes”.

In this sentence: what is due to the structural and biological diversity of terpenes? I can only think that the intention of this sentence may have been “...as a collective representation of the structural and biological diversity of terpenes”.

Answer: These sentences have been corrected.

(Comment 2)

Further to comment (1), there are some phrases used in the text that are vague or ambiguous and should be written with more information, or greater clarity, so that the reader is better informed. For example, the first sentence of the abstract reads “Drug-resistance resulting from efflux pumps is prevalent in clinical Candida isolates, highlighting the efforts of developing novel antifungal agents”. It is of course acceptable that the first sentence of the abstract is general, but it must be informative and clear. This sentence should inform the reader more clearly that it is the expression (or upregulation) of efflux pumps in clinical Candida isolates that has led to the development of drug resistance and that there is a need for new antifungal agents to overcome this.

Answer: We have corrected this sentence in the revised manuscript.

There are in fact many instances of vague or ambiguous passages in the abstract. Another example is in line 20: “...will broaden the way to find more antifungal agents”. In this case, the word “way” is simply too ambiguous for any reader to understand the authors’ intention. Similar instances throughout the manuscript should be addressed.

Answer: We have corrected this sentence in the revised manuscript.

(Comment 3)

As mentioned earlier, the introductory section of the manuscript omits a detailed overview of literature on: (1) mixed assay screening for antimicrobial compounds; and (2) phosphonium-based approaches to drug delivery. As a result, the current work is presented without proper context, and the reader has little understanding of its true relevance or significance. Perhaps the most confusing

aspect for me was to read line 60, where TPP⁺ is first mentioned, followed by the comment “to our surprise”. There is no prior context to phosphonium-based approaches in the text and this comment led me to believe that the authors have made a unique discovery. Later, within the results section (line 95–98), I found that there is literature on phosphonium-based methods for therapeutic application, and that the authors’ motivation to use phosphine appeared to be well founded (and was indeed by design). Since the authors’ comments here are only brief, I am then led to ask the question:

exactly how novel or significant are the results presented here? The reader should not feel this way when reading a Nature Communications article.

This study needs to be clearly placed in the context of prior literature and its significance must be made obvious to the reader. The overall contextual argument needs to be outlined in the introductory section.

Answer: We have added information about the mixed assay screening and phosphonium-tethered approaches in the introduction section in the revised version.

(Comment 4)

In this manuscript, it is claimed that the antifungal activity of PEPS is “restored” when tethered to a triphenylphosphonium moiety. But, by which mechanism/s are the PEPS-TPP⁺ conjugates fungicidal? The word “restored” may suggest that the TPP⁺ moiety is itself benign, and only acts to localise the terpene ‘warhead’ in the mitochondria. While this may indeed be the case, there is little evidence presented here to convince the reader.

It is demonstrated that methyl triphenylphosphonium bromide has no antifungal activity; however, to the best of my knowledge, this compound is a profoundly poor choice for a negative control. It is my understanding that the physicochemical properties of MePPh₃Br are very different from the PEPS-TPP⁺ conjugates reported here. It would be far more convincing to test a range of terpene-TPP⁺ conjugates that were inactive in the Cdr-deficient strains from the primary screen, yet have a somewhat similar structure to the active PEPS. If it were in fact the case that TPP⁺ ligation has no effect on activity, one would expect a null result. However, if the large terpene(non-PEPS)-TPP⁺ conjugates are fungicidal through a different mechanism, conjugates with an ‘inactive warhead’ may still illicit the desired effect.

Answer: We tested the antifungal activity of a (dansyl)-labeled TPP⁺-conjugate and a newly synthesized fluorescent probe, TPP-RhB. These two TPP⁺-conjugates did not exhibit any antifungal effects, indicating that TPP⁺ conjugation itself has no effect on antifungal activity. These data have been added in Table 1. However, these results do not rule out the possibility that some inactive molecules obtained antifungal activity via TPP⁺ conjugation due to structural modification..

This point of contention is critically relevant to the argument presented in this manuscript, and must be addressed. It may be useful to conduct additional experiments and obtain a better understanding of the mechanisms of fungicidal activity (even at a simplistic level). But, regardless of whether or not more results are obtained, a clear in-text explanation of any important conceptual or experimental limitation is necessary.

Answer: In this study, we observed increased mitochondrial membrane potential (MMP) and ROS generation in *C. albicans* cells treated with IS-2-Pi-TPP. MMP alteration is considered a characteristic feature of the apoptosis pathway. We therefore detected cytochrome c (Cyt c) release from mitochondria and performed Annexin V-FITC and propidium iodide (PI) staining. Compared with control cells, mitochondrial Cyt c levels decreased under IS-2-Pi-TPP treatment, whereas cytosolic levels increased significantly, indicating that IS-2-Pi-TPP induced the release of Cyt c from mitochondria in *C. albicans*. Cyt c release from mitochondria is an important feature of apoptosis. Annexin V-FITC and PI staining revealed that cells treated with IS-2-Pi-TPP underwent late cell apoptosis or necrosis, consistent with the alterations of the nucleus revealed by DAPI staining. These results suggest that IS-2-Pi-TPP displays fungicidal activity by inducing both cell apoptosis and necrosis. We have included the new results in the revised version.

(Comment 5)

Finally, some of the images (especially figures 1 and 2) are small and poorly resolved such that they are very difficult to read. This may have been caused through generating the pdf for review, but it may also be an issue for publication. These images should be checked for appropriate size and quality.

Answer: We have increased the quality of the images in the revised version.

REVIEWERS' COMMENTS:

Reviewer #1 (Remarks to the Author):

This is a revised manuscript. I was reviewer #1 for the original manuscript. The authors have addressed most of my concerns. However, It still remains unclear to me as to whether the authors are proposing that the TPP moiety can be used to overcome the efflux of drugs that target cell components other than the mitochondria. The manuscript describes elegant experiments that demonstrate that the TPP-conjugates are not effluxed but are directed to the mitochondria where they exert their effect. So TPP can be used to direct drugs to the mitochondria but can it facilitate the uptake of drugs that target other cellular compartments and allow them to interact with their (non-mitochondrial) targets? The authors show that conjugating TPP to Rhodamine B results in the retention of the conjugate in cells expressing efflux pumps, but the important question is whether that conjugate remains in the cytoplasm or whether it also is directed to the mitochondria. It should be possible to determine the cellular location of the RhB-TPP conjugate with confocal microscopy. If the RhB-TPP conjugate is located in the cytoplasm then it is possible that TPP-conjugates could interact with non-mitochondrial targets, but if the RhB-TPP is located in the mitochondria, then TPP may only be useful for directing drugs to the mitochondria.

The manuscript has been modified extensively. This has introduced some text that needs attention as indicated below:

1. Lines 25-26, what do the authors mean by "the permeability barrier" do they mean the plasma membrane and/or the mitochondrial membrane(s)?
2. Lines 26-27, I think, to be more precise, "thereby combating antifungal resistance" should be replaced with 'thereby combating efflux-mediated antifungal resistance'.
3. Line 76, again, what do the authors mean by "pass through the permeability barrier"?
4. Line 233, it would be helpful if the authors would indicate that the "sub-lethal concentration" of the tested compounds was 0.5 x MIC without having to search through the supplemental material. So the text could be 'sub-lethal concentration (0.5 x MIC)'.
5. Line 260-261, superficial candidiasis does not always result in chronic mucocutaneous candidiasis (CMC). Indeed CMC is a very rare disorder.
6. Line 267, an efficacy cannot be "infinitely superior". 'Superior' will suffice.
7. Lines 301-302, it is unclear what the authors mean by "This characteristic of TPP+ conjugation does not depend on the targets of the parent compounds". The reference cited concerns delivery of TPP conjugates to the mitochondria, so it cannot be used to indicate that TPP could be used to overcome efflux for compounds interacting with other targets.

Reviewer #2 (Remarks to the Author):

[No further comments for author.]

Reviewer #3 (Remarks to the Author):

The authors have suitably addressed all of my prior concerns, and the quality of the revised manuscript is excellent. The work is presented clearly in the context of prior literature and the outcomes of this study are clearly stated with sufficient commentary to describe its novelty and significance. In my opinion, the manuscript in its revised form is suitable for publication after considering my additional comments below, which are limited to minor suggestions/corrections to the text.

Additional comment 1

The first sentence of the abstract (page 2, line 1) has ambiguity in its reference to “novel strategies”. It will be beneficial to the readership if this ambiguity was removed. The following sentence is an example of how the authors might approach removing this ambiguity: “Drug resistance due to upregulation of efflux pumps is prevalent in clinical *Candida* isolates, emphasizing the need to explore novel strategies to overcome this mechanism and to develop new antifungal agents.”

Furthermore, the phrase “...expand opportunities to discover antifungal agents” (page 2, line 5) could be phrased with greater clarity as “...expand the current scope of antifungal hits identified through screening assays”, or something similar.

Additional comment 2

Page 2, line 10, replace the word “was” with “were”.

Additional comment 3

Page 4, line 5-7, please break this sentence into two, or begin with the word “although” and change punctuation to suit. For example, “Although these strategies are associated with some disadvantages such as lower selectivity and reduced antifungal activity, developing alternative strategies of avoiding efflux by pumps would expand opportunities for discovering antifungal agents from natural products.” The authors might also consider my earlier comment regarding clarity of the phrase “...expand opportunities to discover antifungal agents”, which is repeated in this passage.

REVIEWERS' COMMENTS:

Reviewer #1 (Remarks to the Author):

This is a revised manuscript. I was reviewer #1 for the original manuscript. The authors have addressed most of my concerns. However, it still remains unclear to me as to whether the authors are proposing that the TPP moiety can be used to overcome the efflux of drugs that target cell components other than the mitochondria. The manuscript describes elegant experiments that demonstrate that the TPP-conjugates are not effluxed but are directed to the mitochondria where they exert their effect. So TPP can be used to direct drugs to the mitochondria but can it facilitate the uptake of drugs that target other cellular compartments and allow them to interact with their (non-mitochondrial) targets? The authors show that conjugating TPP to Rhodamine B results in the retention of the conjugate in cells expressing efflux pumps, but the important question is whether that conjugate remains in the cytoplasm or whether it also is directed to the mitochondria. It should be possible to determine the cellular location of the RhB-TPP conjugate with confocal microscopy. If the RhB-TPP conjugate is located in the cytoplasm then it is possible that TPP-conjugates could interact with non-mitochondrial targets, but if the RhB-TPP is located in the mitochondria, then TPP may only be useful for directing drugs to the mitochondria.

Response: The intracellular distribution results demonstrated that an average of 52-78% of our tested TPP-conjugates accumulated into mitochondria. This suggests that TPP-conjugates are apt to accumulate into mitochondria but can still arrive in other cellular components with a minor proportion. Therefore, for those potential efflux pump substrates (PEPSS) targeting mitochondria, it is particularly useful to restore or enhance their antifungal activity through chemical conjugation with TPP. For those PEPSS with targeting other intracellular components, maybe it is not the best choice to deliver them to mitochondria through TPP conjugation although this method can increase their intracellular concentration. At present, we have no direct evidence to prove whether TPP-conjugates can interact with non-mitochondrial targets. According to your suggestion, we will further utilize confocal microscopy to observe the intracellular distribution of RhB-TPP and collect more substrates of efflux pumps and determine the intracellular distribution of their TPP-conjugates. More data obtained in the future may clearly and definitely answer your question.

The manuscript has been modified extensively. This has introduced some text that needs attention as indicated below:

1. Lines 25-26, what do the authors mean by “the permeability barrier” do they mean the plasma membrane and/or the mitochondrial membrane(s)?

Response: In lines 25-26, “the permeability barrier” means the plasma membrane barrier. However, the abstract has been re-edited and the permeability barrier has been deleted. We have defined it more specific in the last paragraph of the introduction section.

2. Lines 26-27, I think, to be more precise, “thereby combating antifungal resistance” should be replaced with ‘thereby combating efflux-mediated antifungal resistance’.

Response: Thanks for your suggestion. The abstract has been re-edited and the sentence you mentioned has been removed.

3. Line 76, again, what do the authors mean by “pass through the permeability barrier”?

Response: “The permeability barrier” means the plasma membrane barrier. We have explained it in the introduction section of the revised version.

4. Line 233, it would be helpful if the authors would indicate that the “sub-lethal concentration” of the tested compounds was 0.5 x MIC without having to search through the supplemental material. So the text could be ‘sub-lethal concentration (0.5 x MIC)’.

Response: We have made the requested change in revised manuscript.

5. Line 260-261, superficial candidiasis does not always result in chronic mucocutaneous candidiasis (CMC). Indeed CMC is a very rare disorder.

Response: Thanks for your suggestion. We have changed this sentence from affirmative mood to possible tone.

6. Line 267, an efficacy cannot be “infinitely superior”. ‘Superior’ will suffice.

Response: We have made the requested change in revised manuscript.

7. Lines 301-302, it is unclear what the authors mean by “This characteristic of TPP+ conjugation does not depend on the targets of the parent compounds”. The reference cited concerns delivery of TPP conjugates to the mitochondria, so it cannot be used to indicate that TPP could be used to overcome efflux for compounds interacting with other targets.

Response: Thanks for your reminding. In the main text of cited reference, authors summarize some

examples that some compounds with targets other than the mitochondria or with unknown targets can be carried by TPP conjugation into cancer cells, even cells with overexpressing efflux pump p-glycoprotein.

Reviewer #2 (Remarks to the Author):

[No further comments for author.]

Reviewer #3 (Remarks to the Author):

The authors have suitably addressed all of my prior concerns, and the quality of the revised manuscript is excellent. The work is presented clearly in the context of prior literature and the outcomes of this study are clearly stated with sufficient commentary to describe its novelty and significance. In my opinion, the manuscript in its revised form is suitable for publication after considering my additional comments below, which are limited to minor suggestions/corrections to the text.

Additional comment 1

The first sentence of the abstract (page 2, line 1) has ambiguity in its reference to “novel strategies”. It will be beneficial to the readership if this ambiguity was removed. The following sentence is an example of how the authors might approach removing this ambiguity: “Drug resistance due to upregulation of efflux pumps is prevalent in clinical *Candida* isolates, emphasizing the need to explore novel strategies to overcome this mechanism and to develop new antifungal agents.”

Response: The abstract has been re-edited to remove this ambiguity.

Furthermore, the phrase “...expand opportunities to discover antifungal agents” (page 2, line 5) could be phrased with greater clarity as “...expand the current scope of antifungal hits identified through screening assays”, or something similar.

Response: Thanks for your suggestion. The abstract has been re-edited and this sentence has been

removed.

Additional comment 2

Page 2, line 10, replace the word “was” with “were”.

Response: Thanks for your reminding. The abstract has been re-edited and the sentence you mentioned has been removed.

Additional comment 3

Page 4, line 5-7, please break this sentence into two, or begin with the word “although” and change punctuation to suit. For example, “Although these strategies are associated with some disadvantages such as lower selectivity and reduced antifungal activity, developing alternative strategies of avoiding efflux by pumps would expand opportunities for discovering antifungal agents from natural products.” The authors might also consider my earlier comment regarding clarity of the phrase “...expand opportunities to discover antifungal agents”, which is repeated in this passage.

Response: We have accepted your suggestion in revised manuscript.